EMBO
Molecular Medicine

# TET3 is a regulator and can be targeted for the intervention of myocardial fibrosis

Chenghao Zhu[1,5], Wenxuan Hong[2,5], Yuwen Zhu[3,5], Yujia Xue[1], Zemin Fang[4], Dingsheng Jiang ⓘ [4✉],
Yong Xu ⓘ [3✉] & Ming Kong ⓘ [1,3✉]

## Abstract

**Cardiac fibrosis contributes to adverse cardiac remodeling and loss of heart function eventually leading to heart failure (HF). Resident cardiac fibroblasts are the principal source of myofibroblasts that produce extracellular matrix proteins to mediate cardiac fibrosis. We report that TET3 depletion in cultured cardiac fibroblasts blocked transition to myofibroblasts in response to different pro-fibrogenic stimuli. Consistently, deletion of TET3 from quiescent or activated fibroblast (myofibroblast) attenuated cardiac fibrosis and rescued heart function in mice. Importantly, a small-molecule TET3-specific degrader Bobcat339 displayed therapeutic potential by mitigating cardiac fibrosis and normalizing heart function when administered post-surgery. Integrated transcriptomic analysis identified the mechanosensor Piezo2 as a downstream target for TET3. Piezo2 inhibition dampened fibroblast activation in vitro and ameliorated cardiac fibrosis in vivo. Mechanistically, Piezo2 promoted fibroblast activation by modulating the activities of mechanosensitive transcription factors. Finally, relevance of TET3 and Piezo2 was verified in heart specimens collected from HF patients. In conclusion, our data demonstrate that TET3 is a pivotal regulator of cardiac fibrosis and can be potentially targeted for the intervention of heart failure.**

**Keywords** Heart Failure; Cardiac Fibrosis; Fibroblast; Myofibroblast; Epigenetics
**Subject Category** Cardiovascular System

## Introduction

Heart failure (HF), defined as a failure to pump blood into circulation by the myocardium due to defective contractile and/or dilative capacity, is one of the deadliest forms of human diseases affecting over 50 million individuals across the globe (Khan et al, 2024). Although advancement in precision medicine and care management has greatly improved quality of life in HF patients, the 5-year survival rate following diagnosis remains stagnantly lower than 50% thus causing significant socioeconomic burdens (Jones et al, 2019). Complex in pathogenesis and heterogeneous in etiology, identified risk factors for HF include senility, hypertension, smoking, ischemic heart disease, and chronic kidney disease (Lawson et al, 2020). HF prevalence is projected to increase to 40% by 2030, largely owing to the global pandemic of metabolic disorders (i.e., Diabetes mellitus and obesity). A common pathological feature observed in HF patients is diffuse myocardial fibrosis, characterized by excessive synthesis and deposition of extracellular matrix (ECM) in the myocardium, which leads to adverse ventricular remodeling, ultimately dampening heart function (Lopez et al, 2021), for which no curative pharmacotherapy is currently available.

Regardless of etiology, cardiac fibrosis is mediated by the ECM-producing myofibroblasts, a specialized population of cells that transiently emerge following injury and then promptly evanesce in the wound healing process (Weber et al, 2013). Once precarious and controversial, the origin of myofibroblasts has been unequivocally delineated recently thanks to sophisticated lineage-tracing techniques: most in the field now subscribe to the notion that myofibroblasts arise predominantly, if not exclusively, from resident cardiac fibroblasts (Tallquist and Molkentin, 2017). As elegantly demonstrated in a seminal study by the Molkentin group, resident fibroblasts, developmentally derived from the epicardium/endocardium, trans-differentiate into myofibroblasts positively labeled by the matricellular protein Periostin (*Postn*) in models of ischemic cardiomyopathy (myocardial infarction) and non-ischemic cardiomyopathy (pressure overload induced by Ang II/PE infusion) (Kanisicak et al, 2016). Consistently, ablation of *Postn*[+] myofibroblasts not only reduces cardiac fibrosis but affords cardioprotection by preserving heart function in mice post-injury (Kanisicak et al, 2016; Kaur et al, 2016). The development and application of single-cell transcriptomic tools to studying cardiac fibrosis, while affirming resident fibroblasts as precursors to myofibroblasts, suggest that myofibroblasts are not a homogenous population but consist of several distinct sub-populations (Gladka et al, 2018; Li et al, 2023; Patrick et al, 2024).

When resident cardiac fibroblasts trans-differentiate into myofibroblasts in response to environmental cues, they undergo

[1]State Key Laboratory of Natural Medicines, Department of Pharmacology, China Pharmaceutical University, Nanjing, China. [2]Department of Cardiology, Affiliated Hospital of Jiangnan University, Wuxi, China. [3]Key Laboratory of Targeted Intervention of Cardiovascular Disease and Collaborative Innovation Center for Cardiovascular Translational Medicine, Department of Pathophysiology, Nanjing Medical University, Nanjing, China. [4]Division of Cardiovascular Surgery, Tongji Hospital, Tongji Medical College, Huazhong University of Science and Technology, Wuhan, China. [5]These authors contributed equally: Chenghao Zhu, Wenxuan Hong, Yuwen Zhu. ✉E-mail: jds@hust.edu.cn; yjxu@njmu.edu.cn; mingkong@cpu.edu.cn

profound morphological and transcriptomic alterations. For instance, myofibroblasts tend to be more migraproliferative, more contractile, and more synthetic than quiescent fibroblasts, which are underpinned by transcriptomic reprogramming orchestrated by a network of transcription regulators. In mammalian cells, transcription events are invariably linked to the epigenetic machinery that includes histone and DNA-modifying enzymes, chromatin remodeling proteins, and non-coding RNAs. Ten-and-eleven translocator (TET) family of dioxygenases, i.e., TET1, TET2, and TET3, are *de facto* DNA demethylases functioning to convert 5-methyl-cytosine to cytosine (Xu and Bochtler, 2020). TET proteins display tissue-specific expression patterns and play important roles in embryogenesis and post-natal life activities (Chen and Zhang, 2020). Recently, Huang and colleagues have reported that TET3 brokers crosstalk between hepatocytes and hepatic stellate cells (HSCs) to promote liver fibrosis (Xu et al, 2020). Because HSCs are the undisputable progenitor to hepatic myofibroblasts, the observation by Xu et al (Xu et al, 2020) prompted us to investigate whether and, if so, how TET3 might contribute to myofibroblast activation in the context of cardiac fibrosis. Our data suggest that TET3 plays an essential role in myofibroblast activation and myocardial fibrosis likely by activating the transcription of mechanosensor Piezo2.

## Results

### TET3 is essentialcicardiac fibrosiation in vitro

Human primary cardiac fibroblasts became activated when exposed to TGF-β treatment, as evidenced by elevated expression of myofibroblast markers and augmented cell proliferation/migration/contraction. TET3 silencing by siRNAs, however, significantly dampened fibroblast activation (Fig. 1A–E). Similarly, Cre-mediated TET3 deletion in murine cardiac fibroblasts isolated from the $Tet3^{f/f}$ mice attenuated induction of myofibroblast markers (Fig. 1F,G) and acceleration of cell proliferation/migration/contraction (Fig. 1H–J) by TGF-β treatment. It was noted that TET3 was also essential for cardiac fibroblast activation stimulated by angiotensin II (Appendix Fig. S1) or endothelin (Appendix Fig. S2).

### TET3 is indispensable for cardiac fibrosis in mice

Because of the highly heterogeneous nature of cardiac fibroblasts, we used previously published single-cell RNA-seq data to determine sub-lineage(s) in which TET3 might be preferentially expressed. Three clusters of cardiac fibroblasts were identified: cluster 1 mediates response to stimuli, cluster 2 is involved in cytoskeletal remodeling, whereas cluster 3 is involved in immune response (Wang et al, 2022); TET3 was primarily expressed by cluster 1 cardiac fibroblasts (Appendix Fig. S3). To verify the essentiality of TET3 in cardiac fibrosis in vivo, the $Tet3^{f/f}$ mice were crossed to the $Col1a2$-Cre$^{ERT2}$ mice to generate fibroblast conditional TET3 knockout mice (TET3$^{\Delta F}$). The TET3$^{\Delta F}$ mice and the control mice were subjected to the TAC procedure to induce cardiac fibrosis (Fig. 2A). Pressure overload induced cardiac hypertrophy was comparable in the TET3$^{\Delta F}$ mice and the control mice as evidenced by heart weight/body weight ratio (Fig. 2B),

heart weight/tibia length ratio (Fig. 2C), interventricular septum thickness (IVSd, Fig. 2D), and inferior posterior wall width in diastole (IVPWd, Fig. 2E). PicroSirius Red (PSR) staining and Masson's Trichrome staining revealed a significant decrease in extracellular matrix deposition in the myocardium of the TET3$^{\Delta F}$ mice compared to the control mice (Fig. 2F). Consistently, myofibroblast markers, measured by qPCR (Fig. 2G), and collagenous tissues, measured by hydroxyproline quantification (Fig. 2H), were downregulated as a result of TET3 deletion. Importantly, amelioration of cardiac fibrosis in the TET3$^{\Delta F}$ mice translated to better recovery of heart function, as measured by left ventricular ejection fraction (Fig. 2I) and fractional shortening (Fig. 2J), following the TAC procedure. In an alternative model wherein the mice developed cardiac fibrosis following the LAD procedure to induce myocardial infarction (Appendix Fig. S4).

To further investigate whether TET3 would be indispensable for the maintenance of myofibroblast phenotype in vivo, the $Tet3^{f/f}$ mice were crossed to the $Postn$-Cre$^{ERT2}$ mice to generate myofibroblast conditional TET3 knockout mice (TET3$^{\Delta MF}$), followed by the TAC procedure to induce cardiac fibrosis and heart failure (Fig. 2K). The TET3$^{\Delta MF}$ mice developed similar levels of cardiac hypertrophy as the control mice (Fig. 2L–O) but displayed a less severe phenotype of cardiac fibrosis as evidenced by histological staining (Fig. 2P), hydroxyproline quantification (Fig. 2Q), and myofibroblast marker expression (Fig. 2R). As a result, an improved post-surgical heart function was recorded in the TET3$^{\Delta MF}$ mice compared to the control mice (Fig. 2S,T). The notion that myofibroblast-specific TET3 deletion may ameliorate cardiac function and protect from heart failure was further verified in the model of ischemic cardiomyopathy (Appendix Fig. S5).

### TET3 inhibition dampens fibroblast activation and attenuates cardiac fibrosis in mice

Bobcat339 is a small-molecule TET3 degrader that has been reported to target TET3 for disease intervention in different settings. Indeed, Bobcat339 treatment decreased protein levels, but not mRNA levels, of TET3 in cardiac fibroblasts without altering TET1/TET2 expression (Appendix Fig. S6). Of note, Bobcat339 treatment significantly down-regulated myofibroblast markers and suppressed migraproliferative/contractile behaviors in both murine (Fig. 3A–E) and human (Appendix Fig. S7) primary cardiac fibroblasts. Next, the efficacy of Bobcat339 as an antifibrotic agent in vivo was examined; C57B/6j mice were subjected to the TAC procedure followed by administration with Bobcat339 (Fig. 3F). Similar to TET3 deletion in fibroblasts/myofibroblasts, TET3 inhibition minimally impacted cardiac hypertrophy as measured by echocardiography (Fig. 3G,H), heart weight/body weight ratios (Fig. 3I), and heart weight/tibia length ratios (Fig. 3J). PSR/Masson's staining (Fig. 3K) and hydroxyproline quantification (Fig. 3L) revealed a massive reduction in myocardial collagenous deposition as a result of Bobcat339 administration. QPCR data confirmed that Bobcat339 administration led to a collective down-regulation of myofibroblast markers in the heart (Fig. 3M). These structural changes caused by Bobcat339 administration appeared to translate into better recovery of heart function (Fig. 3N,O). The potency of Bobcat339 against adverse ventricular remodeling and loss of heart function was verified in the model of myocardial infarction (Appendix Fig. S8).

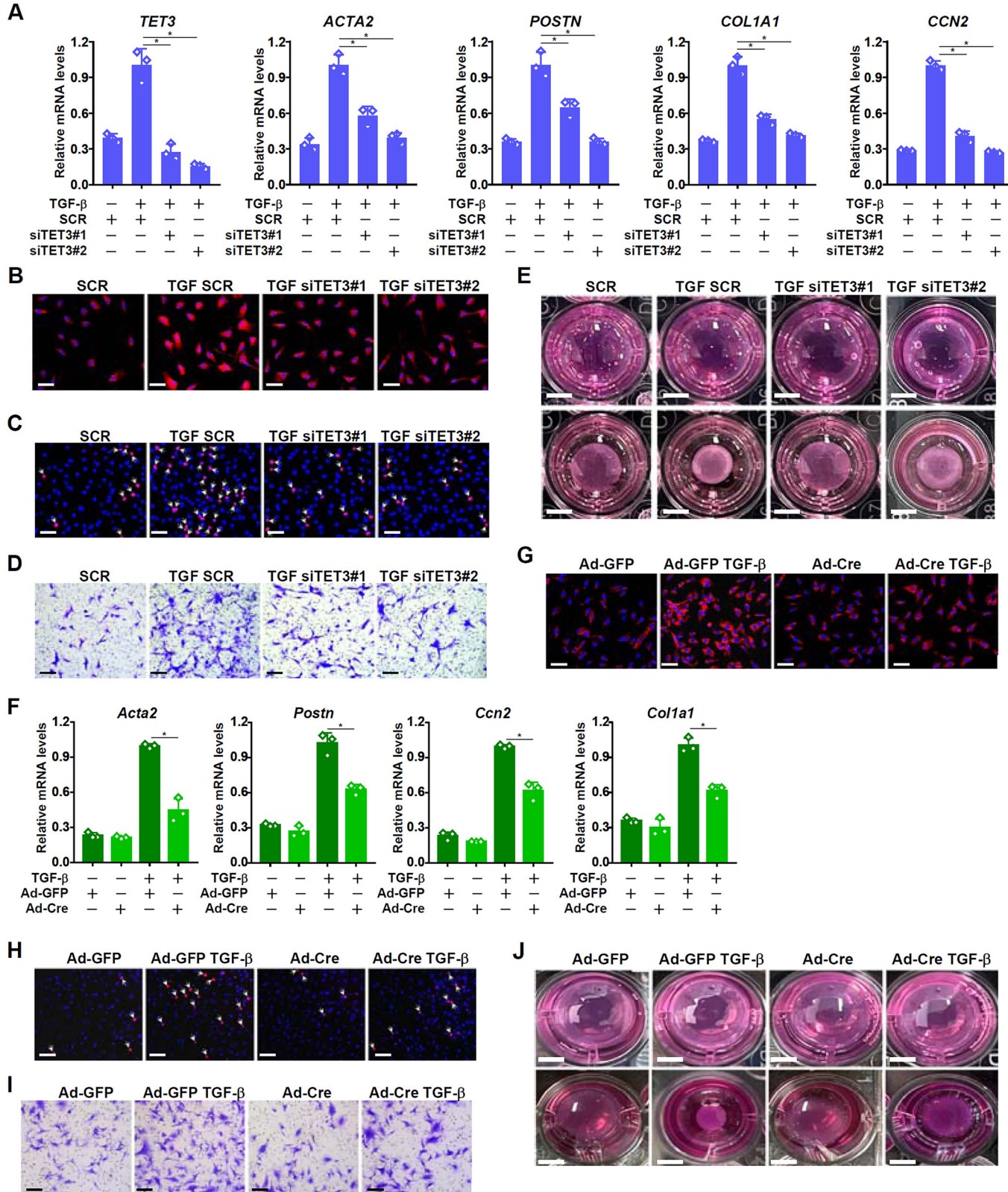

**Figure 1. TET3 is essential for fibroblast activation in vitro.**

(A–E) Human primary cardiac fibroblasts were transfected with siRNAs targeting TET3 or scrambled siRNA (SCR) followed by treatment with TGF-β (5 ng/ml) for 24 h. Myofibroblast markers were examined by qPCR (A). Immunofluorescence staining with an anti-α-SMA antibody. Scale bar, 50 μm (B). EdU incorporation. Scale bar, 50 μm (C). Transwell assay. Scale bar, 50 μm (D). Collagen contraction assay. Scale bar, 1 cm (E). $N = 3$ biological replicates. Data were expressed as mean ± SD. *$p < 0.05$, one-way ANOVA with post hoc Scheffe´s test. Exact $p$ values are reported in Appendix Table S4. (F–J) Primary cardiac fibroblasts isolated from Tet3$^{f/f}$ mice were transduced with Ad-Cre or Ad-GFP, followed by treatment with TGF-β (5 ng/ml) for 24 h. Myofibroblast markers were examined by qPCR (F). Immunofluorescence staining with an anti-α-SMA antibody. Scale bar, 50 μm (G). EdU incorporation. Scale bar, 50 μm (H). Transwell assay. Scale bar, 50 μm (I). Collagen contraction assay. Scale bar, 1 cm (J). $N = 3$ biological replicates. Data were expressed as mean ± SD. *$p < 0.05$, one-way ANOVA with post hoc Scheffe´s test. Exact $p$ values are reported in Appendix Table S4. Source data are available online for this figure.

## TET3 regulates Piezo2 transcription in cardiac fibroblasts

An integrated transcriptomic approach was exploited to explore the mechanism whereby TET3 might regulate fibroblast/myofibroblast phenotype. RNA-seq analysis indicated that Cre-mediated TET3 deletion in cardiac fibroblasts markedly altered the cellular transcriptome, resulting in more than 2000 genes being differentially expressed (Fig. 4A,B). GO analysis (Fig. 4C) and KEGG analysis (Fig. 4D) showed that multiple pathways related to fibroblast activation, including cell migration, ECM synthesis, and cytoskeletal remodeling, were influenced by TET3 deletion. HOMER analysis showed that SMAD3 was among the transcription factors whose activities were downregulated by TET3 (Fig. 4E). On the other hand, CUT&Tag-seq analysis with an anti-TET3 antibody detected a plethora of chromatin peaks across the genome with 6.72% of the peaks occupying the promoter regions (Fig. 4F). GO analysis (Fig. 4G) again suggested that genes where TET3 peaks were preferentially detected were involved in cytoskeletal remodeling, cell proliferation, cell migration, and ECM synthesis. KEGG analysis confirmed that multiple pathways related to fibroblast activation, including EGF signaling, TGF signaling, and Wnt signaling, were enriched in TET3 peaks (Fig. 4H). Prominent pro-fibrogenic transcription factors, including SMAD3, TEAD3, and JUNB, seemed to occupy the chromatin regions where TET3 peaks were detected, as indicated by HOMER analysis (Fig. 4I). Overall, 68 potential TET3 targets were identified by this approach, with the criteria being down-regulated in RNA-seq and occupied by TET3. To further narrow down the targets, one additional criterion was introduced: to be considered a TET3 target with relevance in cardiac fibrosis, the gene would need to be up-regulated in the fibroblast lineage revealed by two separate single-cell RNA-seq datasets, one performed with murine hearts following myocardial infarction (GSE247601) and the other performed with human failing hearts (GSE183852). This stricter screening approach led to the discovery of two putative TET3 targets: Piezo type mechanosensitive ion channel component 2 (Piezo2) and Bicaudal c homolog 1 (Bicc1) (Fig. 4J,K).

Of the two potential TET3 targets, Piezo2 seemed more appealing because it had been demonstrated to be a key mechanosensor and implicated in dermal fibrosis/scarring (Griffin et al, 2023; Pardo-Pastor et al, 2018). A positive correlation between Piezo2 expression in cardiac fibroblasts and myofibroblast marker expression, whereas an inverse correlation between Piezo2 expression in cardiac fibroblasts and heart function was identified in humans by single-cell RNA-seq (Fig. 4L). Piezo2 expression in cultured cardiac fibroblasts was responsive to different pro-fibrogenic stimuli (Appendix Fig. S9). TET3 silencing completely abrogated Piezo2 induction by these stimuli in murine (Fig. 4M,N; Appendix Fig. S10) and human (Fig. 4O,P) cardiac fibroblasts. Piezo2 expression was also down-regulated in the TET3$^{ΔF}$ hearts (Appendix Fig. S11A,B) and TET3$^{ΔMF}$ hearts (Appendix Fig.

S11C,D) compared to the control hearts. Bobcat339 treatment likewise down-regulated Piezo2 expression both in vitro (Fig. 4Q–T) and in vivo (Appendix Fig. S11E,F). ChIP assays showed that TET3 was recruited to the *Piezo2* promoter in response to TGF-β treatment and that TET3 deletion abrogated the accumulation of 5'-hydroxymethyl-cytosine on the *Piezo2* promoter (Fig. 4U,V).

## Piezo2 inhibition ameliorates cardiac fibrosis

To establish a causal relationship between Piezo2 and fibroblast activation, endogenous Piezo2 was silenced in human cardiac fibroblasts by siRNAs; Pieoz2 knockdown markedly retarded fibroblast-myofibroblast transition induced by TGF-β treatment (Appendix Fig. S12). Next, exon 5 of the *Piezo2* allele was floxed to generate the *Piezo2*$^{f/f}$ mice (Appendix Fig. S13A). Cre-mediated Piezo2 deletion in primary cardiac fibroblasts isolated from these mice dampened myofibroblast maturation (Appendix Fig. S13B–E).

The translational potential of targeting Piezo2 in HF intervention was examined by harnessing a small-molecule Piezo2 inhibitor (GsTMx4, Piezo2i). In murine (Fig. 5A–E) and human (Appendix Fig. S14) cardiac fibroblasts, Piezo2i treatment dose-dependently attenuated fibroblast activation. Piezo2i was then administered to mice following the TAC procedure as a therapeutic reagent (Fig. 5F). Piezo2i administration did not influence the course of cardiac hypertrophy (Fig. 5G–J). However, cardiac fibrosis, as measured by PSR/Masson's staining (Fig. 5K), cardiac hydroxyproline levels (Fig. 5L), and expression of myofibroblast markers (Fig. 5M), were significantly alleviated by Piezo2 inhibition. Consequently, the decline of heart function in the compensatory stage was partially averted by Pieoz2i (Fig. 5N,O). Likewise, Pieoz2i administration following the acute phase of myocardial infarction mitigated adverse ventricular remodeling and rescued heart function (Appendix Fig. S15).

## Piezo2 regulates the activity of mechanosensing transcription factors

Integrated transcriptomic assays were performed to gain insights into the mechanism whereby Piezo2 might regulate fibroblast activation. ATAC-seq showed that Piezo2 knockdown lessened chromatin accessibility in cardiac fibroblasts (Fig. 6A). GO (Fig. 6B) and KEGG (Fig. 6C) revealed a similar set of pathways, mostly related to cellular migraproliferative/synthetic behaviors and response to TGF-β, encompassing the genes with altered chromatin accessibility. Geneset enrichment analysis suggested an inverse correlation between Piezo2 deficiency and processes involved in fibroblast-myofibroblast transition (Fig. 6D). HOMER analysis again pointed to a dampening effect of Piezo2 depletion on

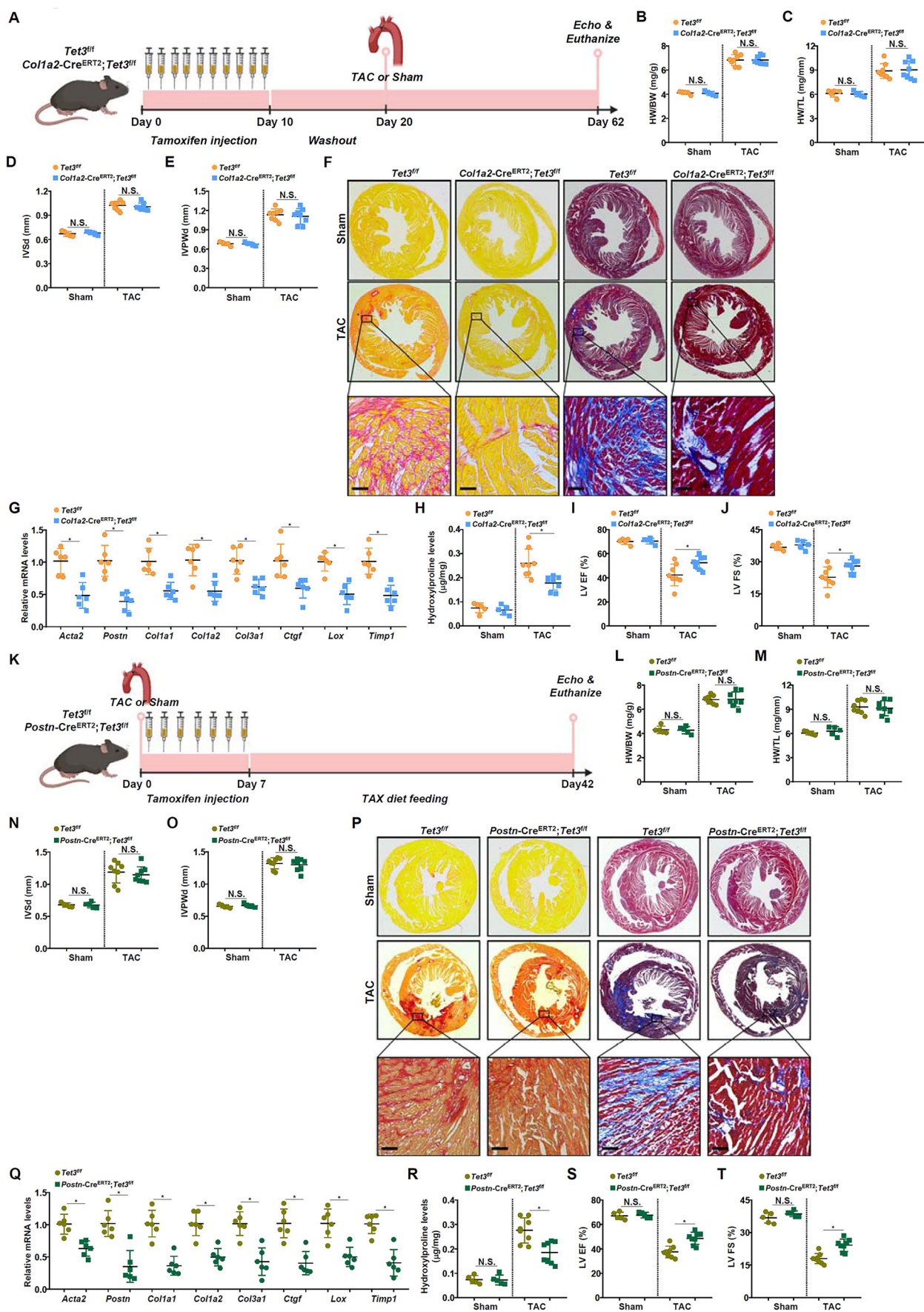

Figure 2. TET3 is indispensable for cardiac fibrosis in mice.

(A–J) Fibroblast conditional TET3 knockout and wild-type mice were subjected to the TAC procedure and euthanized 6 weeks after the surgery. Scheme of protocol (A). Heart weight versus body weight ratio (B). Heart weight versus tibia length ratio (C). IVSd (D). IVWPd (E). Paraffin sections were stained with PicroSirius Red or Masson's Trichrome (F). Myofibroblast markers were examined by qPCR (G). Hydroxyproline levels (H). LV EF (I). LV FS (J). $N = 5$ mice for the sham groups and $N = 8$ mice for the TAC groups. Scale bar, 50 µm. Data were expressed as mean ± SD. *$p < 0.05$, two-tailed Student's test. Exact $p$ values are reported in Appendix Table S4. (K–T) Myofibroblast conditional TET3 knockout and wild-type mice were subjected to the TAC procedure and euthanized 6 weeks after the surgery. Scheme of protocol (K). Heart weight versus body weight ratio (L). Heart weight versus tibia length ratio (M). IVSd (N). IVWPd (O). Paraffin sections were stained with PicroSirius Red or Masson's Trichrome (P). Myofibroblast markers were examined by qPCR (Q). Hydroxyproline levels (R). LV EF (S). LV FS (T). $N = 5$ mice for the sham groups and $N = 8$ mice for the TAC groups. Scale bar, 50 µm. Data were expressed as mean ± SD *$p < 0.05$, two-tailed Student's test. Exact $p$ values are reported in Appendix Table S4. Source data are available online for this figure.

pro-fibrogenic transcription factors such as SRF, PU.1, and TEAD (Fig. 6E). Because SRF and TEAD are well-documented mechanosensitive pro-fibrogenic transcription factors, we therefore performed CUT&Tag-seq to determine the essentiality of Piezo2 in relaying the pro-fibrogenic signal to these factors. As shown in Fig. 6F, enhanced association of SRF and TAZ (proxy for TEAD) with the chromatin in response to TGF-β treatment was markedly weakened in the absence of Piezo2. When the set of genes, whose occupancies by SRF/TEAD were up-regulated by TGF-β but down-regulated by Piezo2 deletion, were annotated, it was discovered that SRF and TAZ have both common and bifurcated targets (Piezo2-dependent) that might regulate various aspects of fibroblast activation: whereas both SRF and TAZ contributed to the regulation of cellular proliferation, cellular contraction, cellular metabolism, and cytoskeletal remodeling, SRF activity appeared to be more aligned with the synthetic ability of activated fibroblasts and TAZ activity appeared to be required for cellular proliferation (Fig. 6G). Together, these data illustrate a potential working model wherein Pieoz2 relays the pro-fibrogenic signal by delegating specific transcriptional functions to multiple mechanosensitive transcription factors.

## Relevance of TET3 and Pieoz2 in humans

Finally, the relevance of TET3 and/or Piezo2 in patients with heart failure was explored. In a small cohort, higher TET3 expression and higher Piezo2 expression were detected in the heart tissues collected from HF patients than from healthy individuals along with higher myofibroblast markers (Fig. 7A). Of note, a positive correlation was identified between TET3 expression and Pieoz2 expression in the human hearts (Fig. 7B). In addition, both TET3 and Piezo2 were positively correlated with myofibroblast markers (Fig. 7C). More importantly, TET3 and Piezo2 appeared to predict worsened heart function (Fig. 7D). Together, these data point to the diagnostic values of TET3 and Piezo2 in heart failure.

## Discussion

Diffuse myocardial fibrosis induced adverse ventricular remodeling is considered a paradigm in HF pathogenesis. Resident cardiac fibroblasts, transitioning from a quiescent state to an activated state (myofibroblasts), are the chief mediator of cardiac fibrosis. Here we detail the involvement of TET3 in orchestrating fibroblast activation and myocardial fibrosis. Consistent with a previous report by the Huang group demonstrating that TET3 activates the TGF-β signaling pathway to promote myofibroblast maturation

(Xu et al, 2020), we show here that deletion of TET3 from quiescent or activated fibroblasts (myofibroblasts) is equivalent in dampening cardiac fibrosis in mice in two different models of cardiomyopathy. Of note, a study by Ngreros et al have documented an up-regulation of TET3 in primary fibroblasts isolated from patients with idiopathic pulmonary fibrosis (IPF) exposed to TGF-β treatment(Negreros et al, 2019). On the contrary, Zeisberg and colleagues have, in a two separate studies, have proposed that TET3 might be an antagonizing factor in cardiac fibrosis. In an earlier study, it has been reported that TET3 down-regulation in cultured endothelial cells apparently induces promoter hypermethylation of the *RASAL1* gene and consequently triggers endothelial-mesenchymal transition (EndMT)(Xu et al, 2015). More recently, the same group has observed that TET3 suppresses proliferation of cultured fibroblasts possibly by promoting DNA double-strand break (DSB) repair (Rath et al, 2024). Since EndMT does not substantively contribute to the pool of ECM-producing myofibroblasts and DNA DSB repair does not represent a causal link to fibrosis, the claim by Zeisberg et al that TET3 inhibits, rather than promotes, cardiac fibrosis, remains to be substantiated short of in vivo genetic evidence. Our data supporting a fibroblast-autonomous role for TET3 in cardiac fibrosis do not foreclose the possibility that TET3 in other cell lineages may contribute to the regulation of this process. For instance, Lv et al have recently reported that TET3 over-expression is sufficient to drive differentiation of monocyte-derived macrophages (MDMs) into "disease-associated macrophages" (DAMs) characterized by expanded viability/resistance to apoptosis and expression of pro-inflammatory mediators including IL-1β and IL-6 (Lv et al, 2024). Since both IL-1β (Siamwala et al, 2023) and IL-6 (Wang et al, 2016) possess the ability to reprogram the fate of cardiac fibroblasts to promote cardiac fibrosis, it is conceivable that TET3 may contribute to cardiac fibrosis by modulating the macrophage phenotype. Additionally, a correlation between TET3 and pro-hypertrophic markers has been identified in the heart in a mouse model of diabetic cardiomyopathy (Ciccarone et al, 2019), raising the possibility that TET3 may regulate cardiac fibrosis secondarily to myocardial injury. These lingering issues certainly warrant further attention in future studies.

We show that the small-molecule TET3 degrader Bobcat339, when administered after the onset of myocardial injury, attenuates cardiac fibrosis. Along with findings by the Huang group that Bobcat339 administration seems to alleviate anorexia nervosa (Lv et al, 2023) and endometriosis (Lv et al, 2024) in mice, our data add to the translational potential of targeting TET3 for the intervention of human diseases. Given that Bobcat339 is a degrader instead of an inhibitor per se, an implicit presumption would be that TET3 could

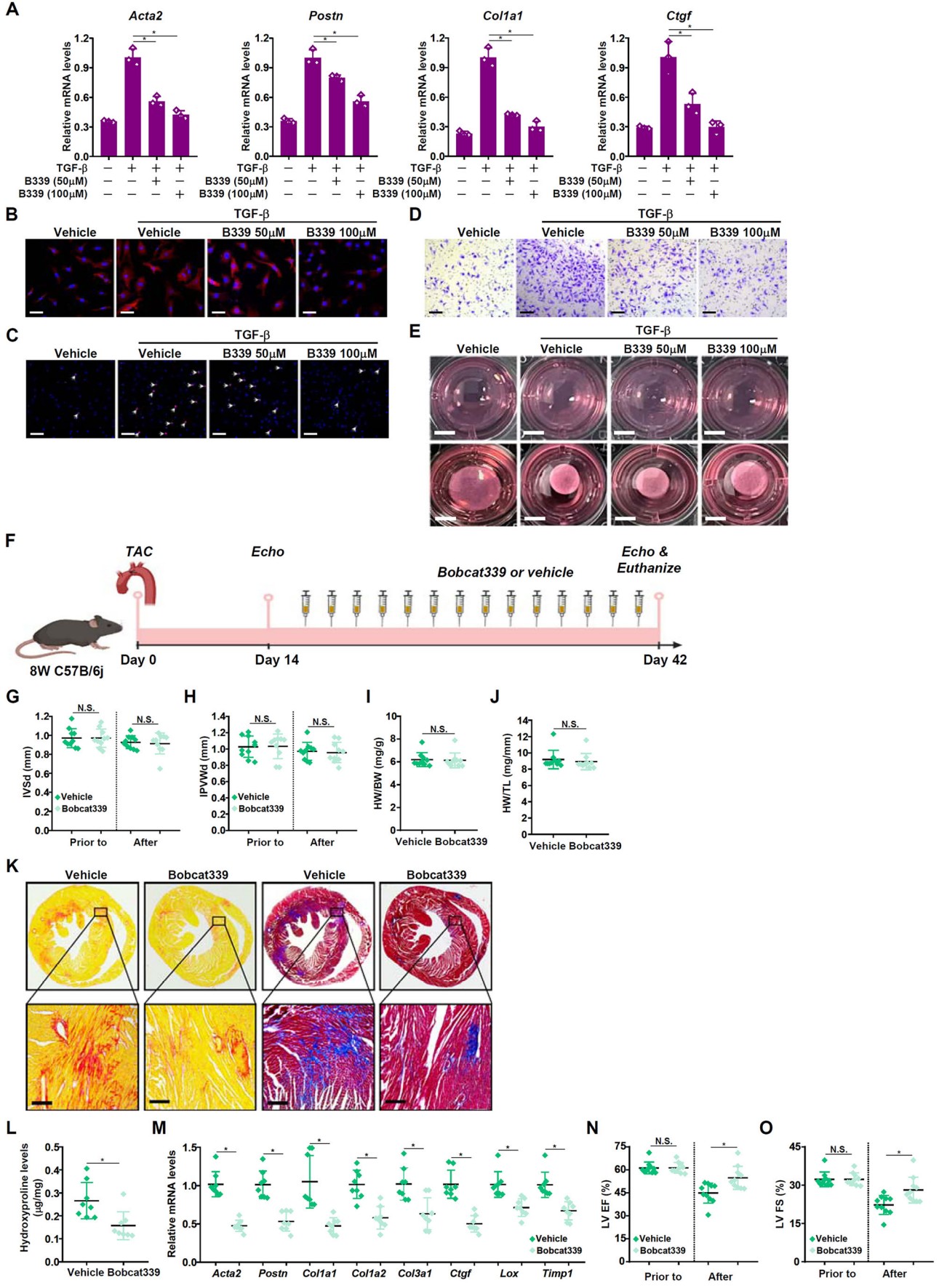

**Figure 3. TET3 inhibition attenuates cardiac fibrosis.**

(A–E) Primary murine cardiac fibroblasts were treated with TGF-β (5 ng/ml) in the presence or absence of Bobcat339 for 24 h. Myofibroblast markers were examined by qPCR (A). Immunofluorescence staining with an anti-α-SMA antibody. Scale bar, 50 μm (B). EdU incorporation. Scale bar, 50 μm (C). Transwell assay. Scale bar, 50 μm (D). Collagen contraction assay. Scale bar, 1 cm (E). $N = 3$ biological replicates. Data are expressed as mean ± SD. *$p < 0.05$, one-way ANOVA with post hoc Scheffe's test. Exact $p$ values are reported in Appendix Table S4. (F–O) C57B/6j mice were subjected to the TAC procedure to induce heart failure followed by intervention with Bocat339. Scheme of protocol (F). Heart weight versus body weight ratio (G). Heart weight versus tibia bone length (H). Interventricular septum thickness (I). Inferior posterior wall width in diastole (J). PicroSirius Red staining and Masson's staining. Scale bar, 50 μm (K). Hydroxyproline levels (L). Myofibroblast markers were examined by qPCR (M). LV EF (N). LV FS (O). $N = 6$–8 mice for each group. Data were expressed as mean ± SD. *$p < 0.05$, two-tailed Student's test. Exact $p$ values are reported in Appendix Table S4. Source data are available online for this figure.

potentially regulate fibroblast activation and, by extension, cardiac fibrosis via catalytic-dependent and catalytic-independent mechanisms. For instance, it has recently been shown that catalytic-independent recruitment of the co-repressors mSin3A and HDAC1 enables TET3 to repress the transcription of Ifnb1 (encoding IFN-β) (Xue et al, 2016). Because a plethora of evidence supports the notion that IFN-β can suppress tissue fibrosis, it is reasonable to speculate that the observed attenuation of cardiac fibrosis in TET3-deficient mice might be attributed to unleashed IFN-β production. More studies, harnessing new mouse models (e.g., TET3 catalytic-deficient mice as described by Ketchum et al (Ketchum et al, 2024)) and genomewide profiling of 5-$^{m}$C/5-$^{hm}$C status, would help differentiate these roles and provide further clarification for the mode of action of TET3.

Through integrated transcriptomic screening and validation, the mechanosensitive ion channel protein Piezo2 is identified as a bona fide target for TET3. Importantly, genetic and pharmaceutical manipulation of Piezo2 achieves similar antifibrotic effects as that of TET3. As this manuscript was under preparation, a report by the Tao group was published essentially portraying Piezo2 as a promoter of cardiac fibrosis (Ding et al, 2024). Although a plurality of previous research focuses on the role of Piezo2 in the nervous system, mounting evidence has pointed to a pivotal role for Piezo2 in regulating the fibrogenic response. Independent studies have found that Piezo2 expression can be used to identify FOXD1$^{+}$ or PDGFRB$^{+}$ mesenchymal cells, both considered precursors to myofibroblasts, in the kidneys (Mochida et al, 2022; Ochiai et al, 2023). The Longaker group has observed that mice with targeted deletion of either Piezo2 or Piezo1 in mature adipocytes display minimal scarring and improved regenerative healing owing to blockade of adipocyte-fibroblast transition (Griffin et al, 2023). Of intrigue, global deletion of Piezo1 or Piezo2 leads to developmental lethality in mice, indicative of functional non-redundancy, although how Piezo2 and Piezo1 coordinate to regulate fibrogenesis remains obscure. ATAC-seq and CUT&Tag-seq reveal that Piezo2 deficiency leads to selective alteration in chromatin accessibility that can likely be explained by impaired recruitment of mechanosensitive transcription factors SRF and TAZ, but the underlying mechanism is not clear at this point. It has long been known that calcium influx through voltage-gated calcium channels induces contractile gene expression (this signature can be acquired during myofibroblast maturation) through promoting SRF recruitment in a cytoskeletal remodeling-dependent manner (Wamhoff et al, 2004). There is also ample evidence to link calcium influx to YAP/TAZ activation in various pathobiological processes (Khalilimeybodi et al, 2023). The hypothesis that Piezo2-mediated calcium trafficking wires mechanosensitive transcription factors to cellular fibrogenic response is certainly worth pursuing in follow-up studies.

Despite the advances proffered by the present study, there are significant limitations that might dampen its enthusiastic reception. First, because the screening criteria for TET3 targets were set very stringently, there is a high likelihood that many molecules with potential fibro-modulatory effects might have been missed. A more thorough annotation should be performed so that the entire spectrum of germane TET3 targets can be uncovered. Second, although Bobcat339 and D-GsMTx4 are shown here to exert antifibrotic effects presumably through targeting TET3 and Piezo2, respectively, the specificity and in vivo pharmacokinetics of these compounds need to be further explored to facilitate translation. In a similar vein, precisely because of the non-specificity of the Piezo2 inhibitor and the lack of genetic evidence (fibroblast/myofibroblast conditional Piezo2 knockout animals), it remains to be ascertained whether Piezo2 can be considered as a genuine regulator of cardiac fibrosis and heart failure in vivo. Third, the sample size of the human specimens used to validate the relevance of our finding is too small to be meaningful. Authentication with larger cohorts across different disease spectrums would further cement the conclusion. Finally, although the Col1a2-Cre$^{ERT2}$ driver has been widely employed for gene targeting in cardiac fibroblasts, its specificity has recently been questioned (Aguado-Alvaro et al, 2022). Future studies should consider alternative, more specific Cre strains (e.g., Tcf21-Cre$^{MCM}$) to further validate the pro-fibrogenic role of TET3 in quiescent fibroblasts. These shortcomings notwithstanding, our data provide sufficient rationale to continue this line of investigation to more clearly define the mechanistic role of TET3 and Piezo2 in cardiac fibrosis so that translational strategies can be devised to combat heart failure.

## Methods

### Reagents and tools table

| Reagent/resource | Reference or source | Identifier or catalog number |
| --- | --- | --- |
| **Experimental models** | | |
| Col1a2-Cre$^{ERT2}$ | Jackson Lab | #029567 |
| Postn-Cre$^{ERT2}$ | Jackson Lab | #029645 |
| Tet3$^{f/f}$ | Prof. Guoliang Xu | N/A |
| Piezo2$^{f/f}$ | Cyagen | #19559 |
| **Recombinant DNA** | | |
| Ad5-CMV-Cre-3FLAG-EGFP | Genechem | N/A |

| Reagent/resource | Reference or source | Identifier or catalog number |
| --- | --- | --- |
| Ad5-CMV-3FLAG-EGFP | Genechem | N/A |
| **Antibodies** | This study | Appendix Table S2 |
| **Oligonucleotides and other sequence-based reagents** | | |
| PCR primers | This study | Appendix Table S1 |
| siRNA | This study | Appendix Table S1 |
| **Chemicals, enzymes, and other reagents** | | |
| Human cardiac fibroblasts | Lifeline | #FC-0060 |
| Dulbecco's modified Eagle's medium (DMEM) | Gibco | #11965092 |
| Fetal bovine serum (FBS) | Gibco | #10270-106 |
| Opti-MEM | Gibco | #31985070 |
| Trypsin(0.25%) | Gibco | #25200072 |
| Lipofectamine™ RNAiMAX Transfection reagent | Invitrogen | #13778150 |
| Click-iT™ Plus EdU Cell Proliferation Kit for Imaging, Alexa Fluor™ 594 dye | Invitrogen | #C10639 |
| Bobcat339 hydrochloride | Selleck | #S6682 |
| D-GsMTx4 TFA | Targetmol | #T37697L |
| Collagen Type I Rat Tail | Corning | #354236 |
| Corning 35 mm TC-treated Culture Dish | Corning | #430165 |
| Corning 60-mm TC-treated culture dish | Corning | #430166 |
| Corning 100-mm TC-treated culture dish | Corning | #430167 |
| Tissue culture plate-six well | Jetbiofil | #TCP011006 |
| Tissue culture plate-12 well | Jetbiofil | #TCP011012 |
| Tissue culture plate-48 well | Jetbiofil | #TCP011024 |
| Cell lysis buffer for Western and IP | Beyotime | #P0013 |
| Penicillin-Streptomycin solution | Beyotime | #C0222 |
| Hydroxyproline assay kit | Solarbio | #BC0250 |
| Masson's Trichrome Stain Kit | Solarbio | #G1346 |
| Picrosirius Red Stain Kit | Solarbio | #G1472 |
| Recombinant human TGF-beta 1 protein | R&D | #7754-BH |
| Goat anti-rabbit IgG (H + L) cross-adsorbed secondary antibody, Alexa Fluor™ 594 | Thermo Fsher | #A-11012 |
| Thermo Scientific PageRuler Plus | Thermo Fisher | #26620 |
| Pierce BCA Protein Assay Kit | Thermo Fisher | #23225 |
| Phenylmethylsulfonyl fluoride(PMSF) | Thermo Fisher | #36978 |
| Tamxifen | Sigma | #T5648 |
| DMSO | Sigma | #D8418 |
| Hyperactive Universal CUT&Tag Assay Kit for Illumina Pro | Vazyme | #TD904-01 |
| VAHTS DNA Clean Beads | Vazyme | #N411 |
| TruePrep Index Kit V2 for Illumina | Vazyme | #TD202 |
| Hyperactive ATAC-Seq Library Prep Kit for Illumina | Vazyme | #TD711 |

| Reagent/resource | Reference or source | Identifier or catalog number |
| --- | --- | --- |
| FastPure Cell/Tissue Total RNA Isolation Kit | Vazyme | #RC101-01 |
| HiScript II Q RT SuperMix for qPCR | Vazyme | #R222-01 |
| ChamQ SYBR Color qPCR Master Mix | Vazyme | #Q411-02 |
| **Software** | | |
| GraphPad Prism9 | GraphPad Software, Inc | RRID:SCR_002798 |
| ImageJ-Fiji | https://imagej.net/software/fiji/ | RRID:SCR_002285 |
| Adobe Illustrator | Adobe Systems | RRID:SCR_010279 |

## Animals

All animal experiments were reviewed and approved by the Ethics Committee on Humane Treatment of Laboratory Animals of China Pharmaceutical University and were performed in accordance with the ethical standards laid down in the 1964 Declaration of Helsinki and its later amendments. Unless specified, the animals were kept under constant environmental conditions with 12 h light/dark cycles and ad libitum access to food and water. $Tet3^{f/f}$ mice (Gu et al, 2011), $Col1a2$-Cre$^{ERT2}$ mice (Lal et al, 2014), and $Postn$-Cre$^{ERT2}$ mice (Kanisicak et al, 2016) have been described previously. $Pieoz2^{f/f}$ mice were generated by inserting LoxP sites to flank exon 5 of the $Pieoz2$ gene. All the strains were on a C57BL/6 J background. To achieve deletion in fibroblasts, the mice were injected with tamoxifen (TAX, 1 mg/25 g BW) for seven consecutive days. To achieve deletion in myofibroblasts, the mice were injected with TAX (1 mg/25 g BW) for 5 consecutive days, followed by feeding with a TAX-containing diet (Teklad, 130860).

Cardiac fibrosis was induced by one of the following methods: (1) Permanent ligation of the left anterior descending coronary artery (LAD) (Yang et al, 2017). Briefly, 8–10 week male mice (25–27 g) were anesthetized with isoflurane and a midline incision was created to expose the heart and the artery was ligated with a 6-0 prolene thread. The wound was then closed using a subcuticular suture. (2) Trans-aortic constriction (TAC) (Liu et al, 2022). Briefly, 8–10 week male mice (25–27 g) were anesthetized, and the hearts were exposed through left thoracotomy in the third intercostal space. A 1.0 mm wire was placed alongside the transverse aorta, which was tied to the wire between the first and second branches of the aortic arch. The wire was quickly removed, leaving the aortic arch constricted to the diameter of the wire. For each type of surgical procedure, the mice were allowed to recover on a heating pad before being returned to the cage. In certain experiments, Bobcat339 (10 mg/kg, MCE, HY-111558) and Piezo2i (10 mg/kg, MCE, HY-P1410B) were injected intraperitoneally (i.p.) every other day following the TAC surgery.

## Histology

Histological analyses were performed essentially as described before (Li et al, 2021). Pictures were taken using an Olympus IX-70 microscope. Quantifications were performed with ImageJ. For each

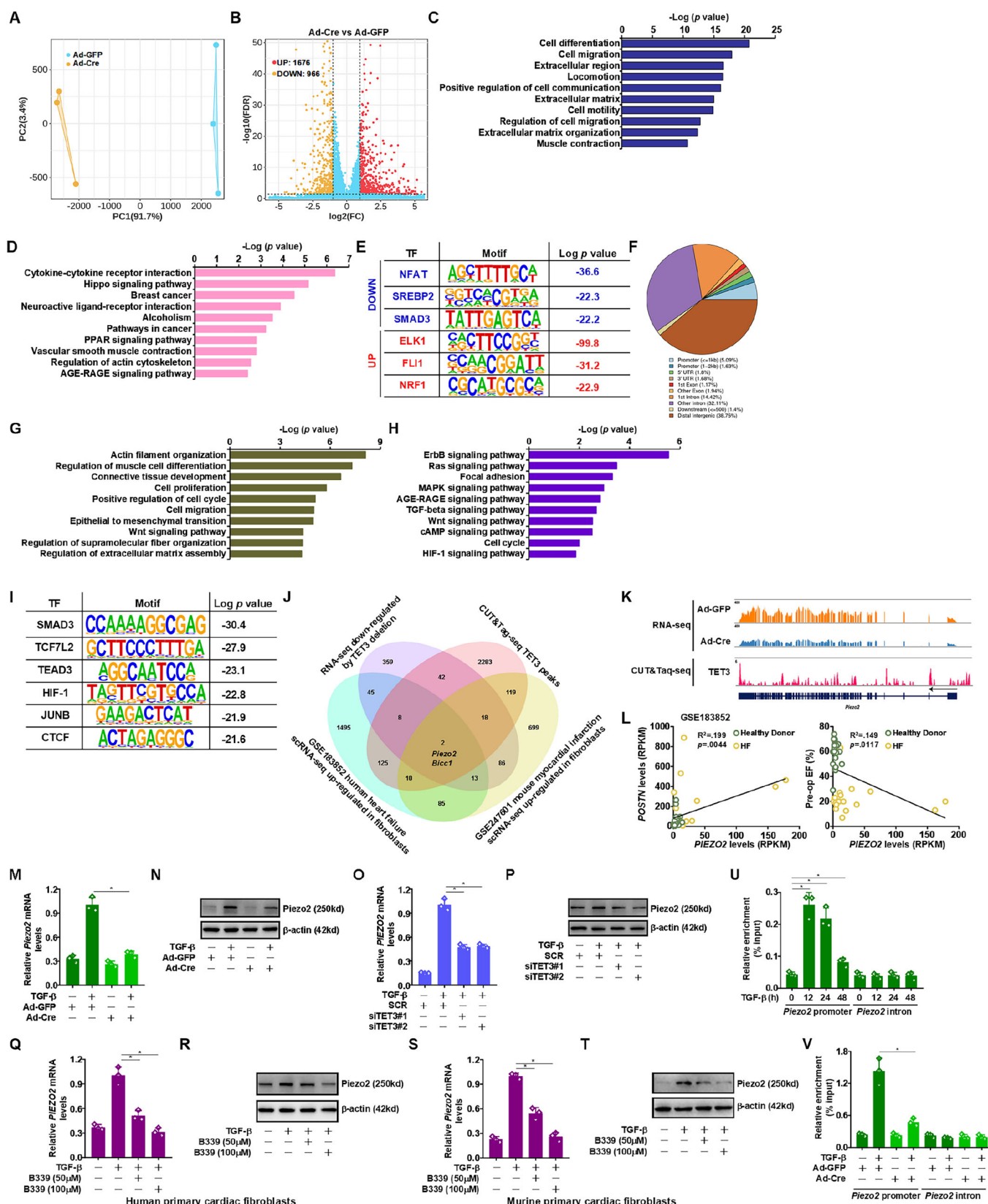

**Figure 4. TET3 regulates Piezo2 transcription in cardiac fibroblasts.**

(A–E) Primary cardiac fibroblasts isolated from *Tet3*[f/f] mice were transduced with Ad-Cre or Ad-GFP, followed by treatment with TGF-β (5 ng/ml) for 24 h. RNA-seq was performed as described in Methods. PCA plot (A). Volcano plot (B). GO analysis (C). KEGG analysis (D). HOMER analysis (E). (F–I) Primary cardiac fibroblasts were treated with TGF-β (5 ng/ml) for 24 h. CUT&Taq-seq was performed as described in Methods. Pie chart of genomic distributions of the TET3 peaks (F). GO analysis (G). KEGG analysis (H). HOMER analysis (I). (J) Venn diagram. (K) CUT&Tag tracks of TET3 signals and RNA-Seq tracks of the read coverage surrounding the *Piezo2* loci. (L) Correlation of fibroblast Piezo2 expression, Periostin expression, and heart function in DCM patients. (M, N) Primary cardiac fibroblasts isolated from *Tet3*[f/f] mice were transduced with Ad-Cre or Ad-GFP followed by treatment with TGF-β (5 ng/ml) for 24 h. Piezo2 expression was examined by qPCR and Western blotting. (O, P) Human primary cardiac fibroblasts were transfected with indicated siRNAs followed by treatment with TGF-β (5 ng/ml) for 24 h. Piezo2 expression was examined by qPCR and Western blotting. (Q, R) Human primary cardiac fibroblasts were treated with TGF-β (5 ng/ml) in the presence or absence of Bobcat339 for 24 h. Piezo2 expression was examined by qPCR and Western blotting. (S, T) Murine primary cardiac fibroblasts were treated with TGF-β (5 ng/ml) in the presence or absence of Bobcat339 for 24 h. Piezo2 expression was examined by qPCR and Western blotting. (U) Murine primary cardiac fibroblasts were treated with TGF-β (5 ng/ml) and harvested at the indicated time points. ChIP assays were performed with anti-TET3. (V) Primary cardiac fibroblasts isolated from *Tet3*[f/f] mice were transduced with Ad-Cre or Ad-GFP followed by treatment with TGF-β (5 ng/ml) for 24 h. ChIP assays were performed with anti-5-hydroxymethyl-cytosine. $N = 3$ biological replicates. Data were expressed as mean ± SD. *$p < 0.05$, one-way ANOVA with post hoc Scheffe's test. Exact $p$ values are reported in Appendix Table S4. Source data are available online for this figure.

## Cell culture, plasmids, and transient transfection

Primary murine cardiac fibroblasts were isolated and cultured as previously described (Khalil et al, 2017). Primary human cardiac fibroblasts were purchased from Lonza. Small interfering RNAs were purchased from Dharmacon. Transient transfections were performed with Lipofectamine 2000 or RNAiMax (Thermo Fisher).

## EdU incorporation assay

5-ethynyl-2′-deoxyuridine (EdU) incorporation assay was performed in triplicate wells with a commercially available kit (Thermo Fisher) as previously described (Shao et al, 2021). Briefly, the EdU solution was diluted with the culture media and added to the cells for an incubation period of 2 h at 37 °C. After several washes with 1XPBS, the cells were then fixed with 4% formaldehyde and stained with Alexa Fluor™ 488. The nucleus was counterstained with DAPI. The images were visualized by fluorescence microscopy and analyzed with Image-Pro Plus (Media Cybernetics). For each well, six different fields were randomly chosen, and the positively stained cells were counted and divided by the total number of cells. The average of the six fields for each well was calculated. The averages of each group were then normalized to the averages of the control group. The data are expressed as relative EdU staining compared to the control group, arbitrarily set as 1. All experiments were performed in triplicate wells and repeated three times. One representative experiment was shown in the figures.

## Boyden chamber migration assay

The cells were trypsinized and seeded into Boyden chambers (PET track-etched, 8-μm pores, 24-well format; Becton Dickinson; cat#354597) in serum-free DMEM medium. Complete culture medium containing 10% FBS was added to the lower chamber. The cells migrating from the upper chamber were fixed with 4% paraformaldehyde, stained with 0.1% crystal violet, and counted under a microscope. Cell numbers from five random fields were counted in each well.

## Collagen contraction assay

The cells were trypsinized, mixed with 4x the volume of Collagen Gel Working Solution (Corning; cat#: 354236) and incubated for

1 h at 37 °C. After collagen polymerization, 1.0 mL of culture medium was added atop. The collagen gel size change was measured 24 h later and quantified with Image-Pro Plus. Data were expressed as relative contraction normalized to the control group, arbitrarily set as 1.

## RNA isolation and real-time PCR

RNA was extracted with the RNeasy RNA isolation kit (Qiagen) as previously described (Kong et al, 2021a; Kong et al, 2021b). Reverse transcriptase reactions were performed using a SuperScript First-strand Synthesis System (Invitrogen). Real-time PCR reactions were performed on an ABI Prism 7500 system. The primers are listed in the Appendix Table S1. Ct values of target genes were normalized to the Ct values of house-keeping control gene (18 s, 5′-CGCGGTTCTATTTTGTTGGT-3′ and 5′-TCGTCTTCGAAAC TCCGACT-3′ for both human and mouse genes) using the ΔΔCt method and expressed as relative mRNA expression levels compared to the control group, which is arbitrarily set as 1.

## Protein extraction and Western blot

Whole cell lysates were obtained by re-suspending cell pellets in RIPA buffer (50 mM Tris pH 7.4, 150 mM NaCl, 1% Triton X-100) with freshly added protease and phosphatase inhibitors (Roche) as previously described (Fan et al, 2021). Antibodies used for Western blotting are listed in the Appendix Table S2.

## RNA sequencing and data analysis

RNA-seq was performed and analyzed as previously described (Wu et al, 2021). Total RNA was extracted using the TRIzol reagent according to the manufacturer's protocol. RNA purity and quantification were evaluated using the NanoDrop 2000 spectro-photometer (Thermo Scientific, USA). RNA integrity was assessed using the Agilent 2100 Bioanalyzer (Agilent Technologies, Santa Clara, CA, USA). Then the libraries were constructed using TruSeq Stranded mRNA LT Sample Prep Kit (Illumina, San Diego, CA, USA) according to the manufacturer's instructions and sequenced on an Illumina HiSeq X Ten platform and 150 bp paired-end reads were generated. Raw data (raw reads) of fastq format were first processed using Trimmomatic, and the low-quality reads were removed to obtain the clean reads. The clean reads were mapped to the mouse genome (Mus_musculus.GRCm38.99) using HISAT2.

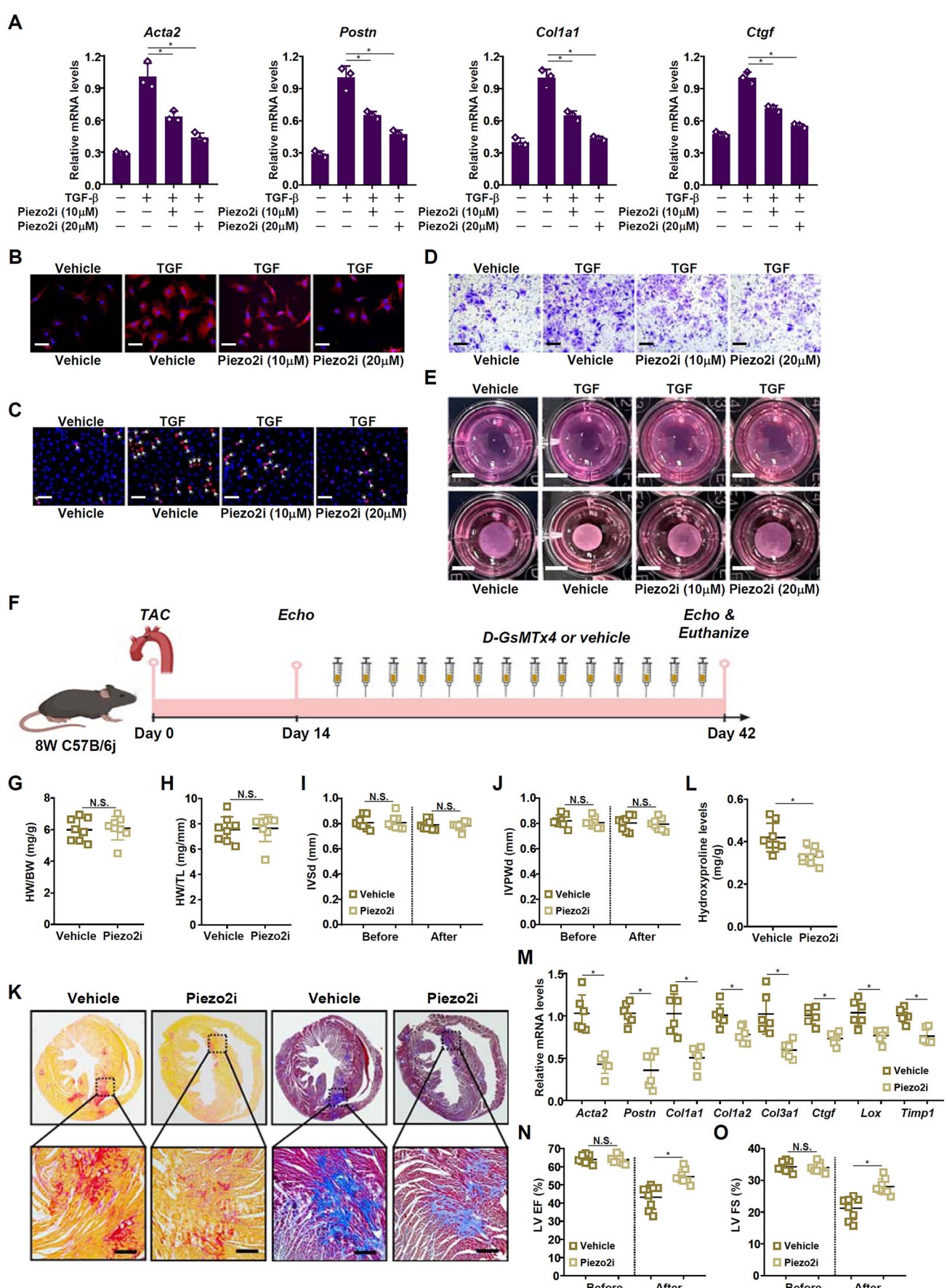

**Figure 5.   Piezo2 inhibition attenuates cardiac fibrosis.**

(A–E) Primary murine cardiac fibroblasts were treated with TGF-β (5 ng/ml) in the presence or absence of Piezo2i for 24 h. Myofibroblast markers were examined by qPCR (A). Immunofluorescence staining with an anti-α-SMA antibody. Scale bar, 50 μm (B). EdU incorporation. Scale bar, 50 μm (C). Transwell assay. Scale bar, 50 μm (D). Collagen contraction assay. Scale bar, 1 cm (E). $N = 3$ biological replicates. Data were expressed as mean ± SD. $*p < 0.05$, one-way ANOVA with post hoc Scheffé's test. Exact $p$ values are reported in Appendix Table S4. (F–O) C57B/6j mice were subjected to the TAC procedure to induce heart failure, followed by intervention with Piezo2i. Scheme of protocol (F). Heart weight versus body weight ratio (G). Heart weight versus tibia bone length (H). Interventricular septum thickness (I). Inferior posterior wall width in diastole (J). PicroSirius Red staining and Masson's staining. Scale bar, 50 μm (K). Hydroxyproline levels (L). Myofibroblast markers were examined by qPCR (M). LV EF (N). LV FS (O). $N = 6$–8 mice for each group. Data were expressed as mean ± SD. $*p < 0.05$, two-tailed Student's test. Exact $p$ values are reported in Appendix Table S4. Source data are available online for this figure.

FPKM of each gene was calculated using Cufflinks, and the read counts of each gene were obtained by HTSeq-count. Differential expression analysis was performed using the DESeq (2012) R package. $P$ value < 0.05 and fold change > 1.5 or fold change < 0.66 was set as the threshold for significantly differential expression. Hierarchical cluster analysis of differentially expressed genes (DEGs) was performed to demonstrate the expression pattern of genes in different groups and samples. GO enrichment and KEGG pathway enrichment analysis of DEGs were performed, respectively, using R based on the hypergeometric distribution.

## ATAC-seq experiment and data processing

ATAC-seq was performed using an Active Motif ATAC-Seq Kit (Active Motif, 53150) according to the manufacturer's instructions. Briefly, 20 mg tissue was minced with a razor blade to a size of 1 mm² in cold PBS, resuspended in 1 ml Lysis Buffer, and transferred to 1 ml dounce homogenizer for homogenization on ice. Tissue lysates were filtered through a 40-μm cell strainer (Falcon, 352340). About 100,000 nuclei were aliquoted and centrifuged at $500 \times g$ at 4 °C for 5 min. The nuclei pellets were resuspended in 50 μl Transposition Master Mix and incubated at 37 °C for 30 min in a thermomixer set at 800 rpm (Eppendorf ThermoMixer C, Eppendorf, Enfield, CT, USA). Transposed DNA was purified with SPRI beads and eluted in 35 μl Elution Buffer. The library was generated using Q5 High-Fidelity DNA Polymerase and sequenced on an Illumina X10 platform with PE150 strategy. Raw sequences were adapter-trimmed and mapped to hg38 (or mm10, etc) using bwa. PCR duplicates were removed using Picard. Peak calling was performed by MACS2 with parameter (f = BAMPE; nomodel; shift = −100; extsize = 00; $q = 0.05$) and annotated by HOMER. Consensus peaks regions were identified between samples by Bedtools and counted intensity by featureCounts. The "DESeq2" package in R was used to identify the differential peaks. Pearson correlation was calculated based on all the peaks by deeptools. Motif discovery was performed by HOMER. TSS/TES/Genebody enrichment was calculated by deeptools. The "ClusterProfiler" package in R was used to perform the Go enrichment and KEGG pathway enrichment of differential peaks.

## CUT&Tag-sequencing and data processing

CUT&Tag assay was performed per vendor recommendations (Vazyme; cat#TD904). For each CUT&Tag experiment, $1 \times 10^5$ cells were incubated with 10 μl of ConA Beads. Primary antibodies were then incubated overnight with the chromatin at 4 °C on a shaking platform. Beads-chromatin-antibody mixture was washed once

with 200 μl of Dig-wash Buffer, resuspended in 50 μl of Dig-wash Buffer with a mouse secondary antibody, and incubated for 30 min at room temperature on a rotator. The mixture was washed twice with 200 μl of Dig-Wash Buffer and resuspended in 100 μl of Dig-300 Buffer containing 2 μl pA/G-Tnp Pro (protein A–Tn5 transposase fusion protein). After incubation with pA/G-Tnp Pro on a rotator at room temperature, the mixture was washed three times with 200 μl Dig-300 Buffer to remove unbound pA/G-Tnp Pro, and resuspended with 50 μl Tagmentation buffer. Subsequently, the reaction was stopped with 2 μl 10% SDS and 1 pg DNA Spike-in. DNA was extracted with phenol–chloroform and ethanol. Libraries were amplified according to the manufacturer's instructions. Library quality was evaluated using agarose gel electrophoresis. The libraries were sequenced on an Illumina HiSeq 2500. Quality-filtered reads were mapped to the reference genome (GRCm39) using Bowtie2. Masc2 was used to call peaks. Deep tools and the integrative genomics viewer (IGV) were used to accomplish data visualization.

## Human heart specimens

Patients with end-stage dilated cardiomyopathy or ischemic cardiomyopathy were evaluated and definitively diagnosed in accordance with the International Society for Heart and Lung Transplantation (ISHLT) guidelines and subsequently underwent heart transplantation at Tongji Hospital, Tongji Medical College, Huazhong University of Science and Technology. The samples were procured from the left ventricles of the hearts during the transplantation procedure. Prior to the collection of samples, written informed consent was obtained from all patients. The collection of samples was conducted in accordance with a human research protocol that had been approved by the Human Research Ethics Committees of Tongji Hospital, Tongji Medical College, Huazhong University of Science and Technology. Patient information is summarized in the Appendix Table S3.

## Statistical analysis

For comparison between two groups, a two-tailed $t$-test was performed. For comparison among three or more groups, one-way ANOVA or two-way ANOVA with post hoc Turkey analyses were performed using the SPSS package. The assumptions of normality were checked using the Shapiro–Wilk test, and equal variance was checked using Levene's test; both were satisfied. $p$ values smaller than 0.05 were considered statistically significant (*). All in vitro experiments were repeated at least three times, and three replicates were estimated to provide 80% power.

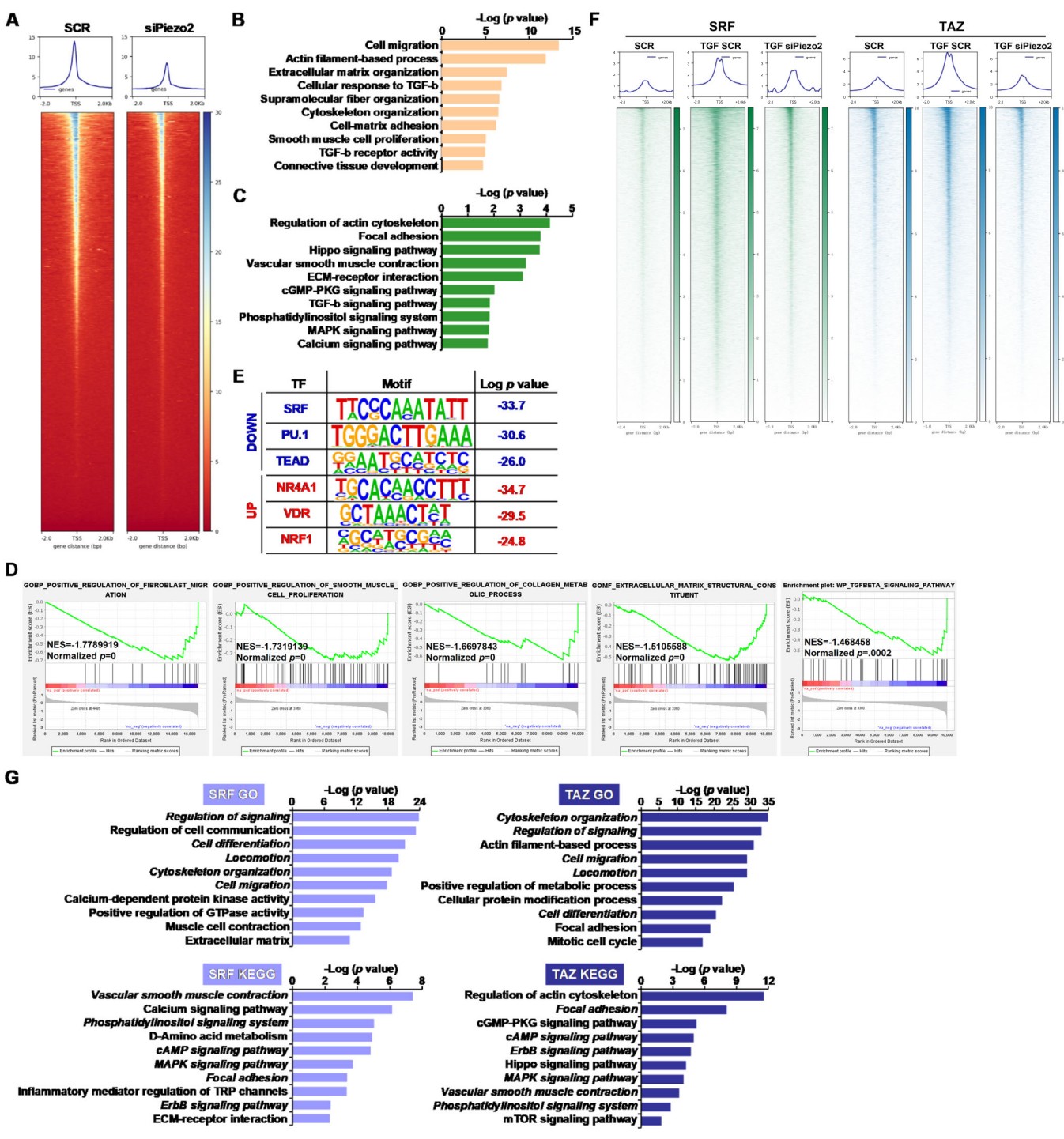

**Figure 6. Piezo2 regulates the activity of mechanosensing transcription factors.**

(A–E) Primary murine cardiac fibroblasts were transfected with siRNA targeting Piezo2 or scrambled siRNA (SCR) followed by treatment with TGF-β (5 ng/ml) for 24 h. ATAC-seq was performed as described in Methods. Heatmap of peaks (A). GO analysis (B). KEGG analysis (C). Statistical significance for GO/KEGG was evaluated through the hypergeometric test using Phyper. Geneset enrichment analysis (D). Statistical significance for GESA was evaluated through sample-label permutation testing (*n* = 1000). HOMER analysis (E). (F, G) Primary murine cardiac fibroblasts were transfected with siRNA targeting Piezo2 or scrambled siRNA (SCR) followed by treatment with TGF-β (5 ng/ml) for 24 h. CUT&Tag-seq was performed as described in Methods. Heatmap of peaks (F). GO and KEGG analysis (G).

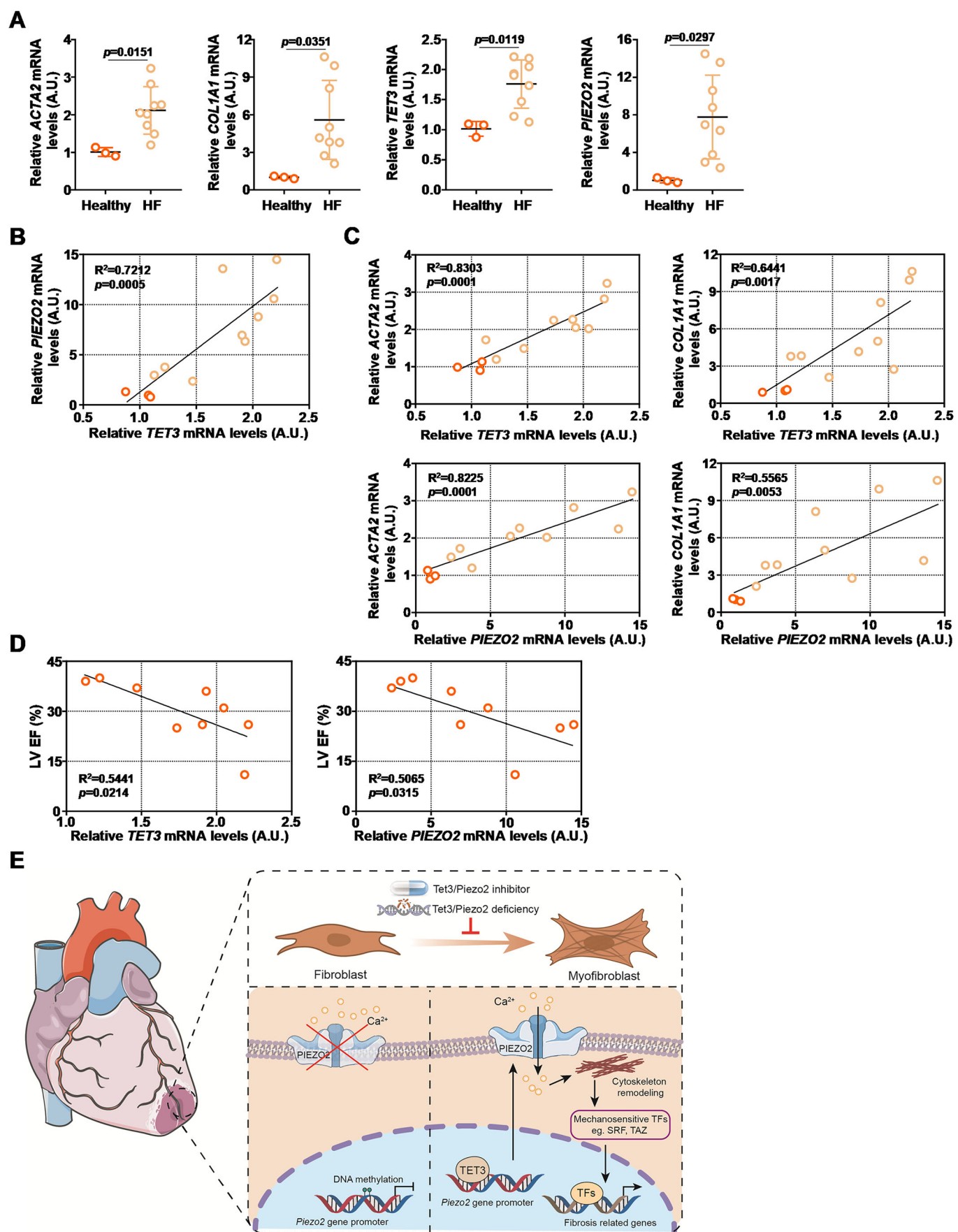

◄ **Figure 7. Relevance of TET3 and Piezo2 in humans.**

(A) Gene expression levels in heart tissues were examined by qPCR. $N = 3$ for healthy individuals and $N = 9$ for HF patients. Data were expressed as mean ± S.D. Two-tailed student's test. (B) Correlation between TET3 expression and Piezo2 expression. (C) Correlation between TET3/Piezo2 and myofibroblast markers. (D) Correlation between TET3/Piezo2 and heart function. (E) A schematic model. Source data are available online for this figure.

## The paper explained

### Problem

Diffuse myocardial fibrosis dampens heart function and leads to heart failure (HF). Trans-differentiation of quiescent cardiac fibroblasts into myofibroblasts is a paradigm in cardiac fibrosis. Effective therapeutic solutions against cardiac fibrosis and HF are limited.

### Results

In this paper Zhu et al describes a novel mechanism whereby the DNA demethylase TET3 contributes to fibroblast-myofibroblast transition and cardiac fibrosis. Genetic and pharmaceutical manipulation of TET3 attenuates cardiac fibrosis and rescues HF. TET3 activates transcription of Piezo2, a mechanosensor, to relay the pro-fibrogenic signals to sequence-specific transcription factors. Importantly, a correlation between TET3, Piezo2, and myofibroblast activation is identified in heart tissues collected from HF patients.

### Impact

These data provide novel insights and translational potential for HF intervention.

## Data availability

RNA-seq data generated for this study have been deposited in the PubMed database with the accession number GSE302976. TET3 CUT&Tag-seq data generated for this study have been deposited in the PubMed database with the accession number GSE302977. Piezo2 ATAC-seq data generated for this study have been deposited in the PubMed database with the accession number GSE305081. SRF/TAZ CUT&Tag-seq data generated for this study have been deposited in the PubMed database with the accession number GSE305080.

The source data of this paper are collected in the following database record: biostudies:S-SCDT-10_1038-S44321-025-00305-4.

## Peer review information

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

## Acknowledgements

This work was supported by grants from the National Natural Science Foundation of China (82121001, 82300298, 82370633, and 82400729).

## Author contributions

**Chenghao Zhu**: Investigation; Methodology; Writing—original draft; Writing—review and editing. **Wenxuan Hong**: Funding acquisition; Investigation; Methodology; Writing—original draft; Writing—review and editing. **Yuwen Zhu**: Investigation; Methodology; Writing—original draft; Writing—review and editing. **Yujia Xue**: Investigation; Methodology; Writing—original draft; Writing—review and editing. **Zemin Fang**: Investigation; Writing—original draft; Writing—review and editing. **Dingsheng Jiang**: Supervision; Funding acquisition; Investigation; Methodology; Writing—original draft; Writing—review and editing. **Yong Xu**: Supervision; Funding acquisition; Writing—original draft; Writing—review and editing. **Ming Kong**: Supervision; Funding acquisition; Investigation; Writing—original draft; Writing—review and editing.

Source data underlying figure panels in this paper may have individual authorship assigned. Where available, figure panel/source data authorship is listed in the following database record: biostudies:S-SCDT-10_1038-S44321-025-00305-4.

## Disclosure and competing interests statement

The authors declare no competing interests.

