## [Peer Review File · EMBO Molecular Medicine]

TET3 is a regulator and can be targeted for the intervention of myocardial fibrosis

Chenghao Zhu, Wenxuan Hong, Yuwen Zhu, Yujia Xue, Zemin Fang, Ding-Sheng Jiang, Yong Xu, and Ming Kong

Corresponding authors: Ming Kong (mingkongcpu22@hotmail.com) , Ding-Sheng Jiang (jds@hust.edu.cn), Yong Xu (yjxu@cpu.edu.cn)

Review Timeline:

Submission Date:	22nd Feb 25
Editorial Decision:	19th Mar 25
Revision Received:	19th Jul 25
Editorial Decision:	7th Aug 25
Revision Received:	11th Aug 25
Accepted:	18th Aug 25

Editor: Jingyi Hou

Transaction Report:

19th Mar 2025

Dear Dr. Kong,

Thank you again for submitting your work to EMBO Molecular Medicine. We have now received feedback from the three referees who agreed to evaluate your manuscript. As you will see in the reports below, the referees find your study of interest but have raised several concerns that will need to be addressed in a major revision of the manuscript.

The referees' recommendations are clear, so I won't repeat the points listed below. It is important to carefully address all the issues raised by the referees. Please feel free to contact me in case you would like to discuss in further detail any of the issues raised by the reviewers.

We would welcome the submission of a revised version within three months for further consideration. As you may already know, our editorial policy allows in principle a single round of major revision, and it is therefore essential to provide responses to the referees' comments that are as complete as possible.

Please also contact us as soon as possible if similar work is published elsewhere. If other work is published, we may not be able to extend the revision period beyond three months.

I look forward to receiving your revised manuscript.

Sincerely,
Jingyi

Jingyi Hou
Senior Editor
EMBO Molecular Medicine

We require:

2) Individual production quality figure files as .eps, .tif, .jpg (one file per figure). For guidance, download the 'Figure Guide PDF': (<https://www.embopress.org/page/journal/17574684/authorguide#figureformat>).

3) A .docx formatted letter INCLUDING the reviewers' reports and your detailed point-by-point responses to their comments. As part of the EMBO Press transparent editorial process, the point-by-point response is part of the Review Process File (RPF), which will be published alongside your paper.

4) A complete author checklist, which you can download from our author guidelines (<https://www.embopress.org/page/journal/17574684/authorguide#submissionofrevisions>). Please insert information in the checklist that is also reflected in the manuscript. The completed author checklist will also be part of the RPF.

6) It is mandatory to include a 'Data Availability' section after the Materials and Methods. Before submitting your revision, primary datasets produced in this study need to be deposited in an appropriate public database, and the accession numbers and database listed under 'Data Availability'. Please remember to provide a reviewer password if the datasets are not yet public (see <https://www.embopress.org/page/journal/17574684/authorguide#dataavailability>).

.

12) Author contributions: You will be asked to provide CRediT (Contributor Role Taxonomy) terms in the submission system. These replace a narrative author contribution section in the manuscript.

13) A Conflict of Interest statement should be provided in the main text.

14) Every published paper now includes a 'Synopsis' to further enhance discoverability. Synopses are displayed on the journal webpage and are freely accessible to all readers. They include a short stand first (maximum of 300 characters, including space)

as well as 2-5 one-sentences bullet points that summarizes the paper. Please write the bullet points to summarize the key NEW findings. They should be designed to be complementary to the abstract - i.e. not repeat the same text. We encourage inclusion of key acronyms and quantitative information (maximum of 30 words / bullet point). Please use the passive voice. Please attach these in a separate file or send them by email, we will incorporate them accordingly.

Please also suggest a visual abstract to illustrate your article as a PNG file 550 px wide x 300-600 px high.

15) All Materials and Methods need to be described in the main text using our 'Structured Methods' format. According to this format, the Methods section includes a Reagents and Tools Table (listing key reagents, experimental models, software and relevant equipment and including their sources and relevant identifiers) followed by a Methods and Protocols section describing the methods, ideally using a step-by-step protocol format. The aim is to facilitate adoption of the methodologies across labs.

Please download and fill our Reagents and Tools Table template (.docx), which you can find in our author guidelines: <https://www.embopress.org/page/journal/17574684/authorguide#structuredmethods>

***** Reviewer's comments *****

Referee #1 (Comments on Novelty/Model System for Author):

This study is interesting. The authors demonstrate the involvement of TET3-Piezo2 axis in the progression of cardiac fibrosis using fibroblast-specific TET3 and Piezo2 deficient mice and already known TET3 degrading small molecule, Bobcat339. They also suggest the involvement of TET3-Piezo2 axis in patients with heart failure. This paper is well organized, and the data are solid and concretely demonstrated.

Referee #1 (Remarks for Author):

This study is interesting. The authors demonstrate the involvement of TET3-Piezo2 axis in the progression of cardiac fibrosis using fibroblast-specific TET3 and Piezo2 deficient mice and already known TET3 degrading small molecule, Bobcat339. They also suggest the involvement of TET3-Piezo2 axis in patients with heart failure. This paper is well organized, and the data are solid and concretely demonstrated. However, there are concerns on data presentations.

Comments:

1. Scale bar should be shown, e.g., Figure 1B-D, G-I.
2. It is curious for me why background color is so high in Ad-GFP TGF- β image compared with other groups. The background color, especially non-cell area, should be the same among 4 groups.
3. Figure 2F, 2P, 3K and 4K. Why do they show only two groups? Is it sham groups or TAC-treated groups? It is sufficient to show either picosirius red images or masson trichrome images, but they need to show all 4 groups.
4. Figure 3. I am wondering if the concentration (50-100 μ M) of Bobcat339 is high. Is this reagent less selective? If so, they need to confirm whether the pharmacological action of Bobcat339 is associated with TET3 degradation.

Referee #2 (Comments on Novelty/Model System for Author):

Authors showed that the TET3-Piezo2 pathway in fibroblasts promotes cardiac fibrosis. In vitro experiments using fibroblasts, and in vivo experiments using fibroblast-specific knockout mice and TET3 degrader (Bobcat339) all strongly indicate the involvement of TET3-Piezo2 in fibrosis. The results obtained are consistent between figures, making the results convincing. However, there is a concern. The heterogeneity has been reported in fibroblasts. It would be better to mention in which type of fibroblasts TET3 is expressed.

Referee #2 (Remarks for Author):

In this study, authors showed that TET3 is involved in fibrosis by promoting the conversion of fibroblasts to myofibroblasts by controlling the expression of Piezo2. In vitro and in vivo experiments have shown consistent results of the importance of the TET3-Piezo2 pathway. In vivo, the effect of Bobcat339, a TET3-specific degrader, was also observed, which is interesting from a clinical side. However, in recent years, heterogeneity has been reported in fibroblasts, so I think it is necessary to mention in which type of cells TET3 is expressed. Overall, they have provided sufficient data, and no further experiments are necessary.

Referee #3 (Comments on Novelty/Model System for Author):

It is not clear whether the Col1a2-CreERT2 mice used in the study had cardiac fibroblast-specific Tet3 ablation.

Referee #3 (Remarks for Author):

RE: EMM-2025-21496

Using siRNA knockdown in mouse and human primary cardiac fibroblasts the authors showed that TET3 was required for fibroblast activation and myofibroblast differentiation induced by profibrotic stimuli such as TGF- β 1. In two different murine models of cardiomyopathy (TAC and LAD), the authors demonstrated that fibroblast (Col1a2-CreERT2)- or myofibroblast (Postn-CreERT2)-specific Tet3 ablation mitigated cardiac fibrosis and improved heart function and that the effects were mimicked by Bobcat339, a small molecule TET3-specific degrader. To investigate the underlying mechanisms, the authors performed Cre-mediated TET3 knockdown in cardiac fibroblasts followed by integrated genome-wide analyses, RNA-seq and CUT&Tag-seq, and identified genes and pathways affected by TET3.

Among the 68 putative TET3 targets identified, the authors focused on Piezo2, a mechanosensitive ion channel protein previously implicated in dermal fibrosis and cardiac fibrosis by other research groups. The authors provided in vitro evidence in cardiac fibroblasts showing that TET3 positively regulated Piezo2 transcription at least in part by binding to the Piezo2 gene promoter inducing 5hmC modification. To address the in vivo role of Piezo2, the authors used a Piezo2 inhibitor, GsTMx4, and observed therapeutic effects in two mouse models of cardiac fibrosis. To find out how Piezo2 might regulate fibroblast activation, the authors performed ATAC-seq and CUT&Tag-seq analyses in cardiac fibroblasts and observed decreased chromatin association of pro-fibrogenic transcription factors SRF and TEAD following Piezo2 knockdown. Finally, the authors revealed a positive correlation between the expression of TET3 and Piezo2 in human heart tissues as well as a higher expression of TET3 and Piezo2 in heart tissues of patients with heart failure as compared to healthy controls. The authors concluded that TET3 plays an important role in the pathogenesis of cardiac fibrosis and may be potentially targeted for therapy.

Overall, the paper was well-written and data were clearly presented. The findings that TET3 in cardiac fibroblasts contributes critically to myocardial fibrosis and that TET3 may serve as a potential therapeutic target are novel and important for the cardiac research field. In addition, the genome-wide data generated from the study provide mechanistic insights onto how TET3 regulates cardiac fibroblasts in the context of myocardial fibrosis, and, if made publicly available, will be a rich source for the research community.

Major points:

1. The importance of the TET3-Piezo2 axis in cardiac fibrosis is not well supported. The authors showed in cardiac fibroblasts that TET3 regulated Piezo2 transcription but did not provide further evidence that Piezo2 is a major downstream effector of TET3, especially given the multitude of potential TET3 targets identified by RNA-seq and CUT&Tag-seq. The authors showed partial therapeutic effects using a Piezo2 inhibitor (GsTMx4). However, as GsTMx4 is not a Piezo2-specific inhibitor and which also inhibits other cationic mechanosensitive channels such as Piezo1 and TRP channels, the in vivo role of Piezo2 as a major downstream effector of TET3 cannot be established. Perhaps, the authors can overexpress TET3 in cardiac fibroblasts using viral vectors (not by TGF- β 1 stimulation which will alter gene expression in addition to expression of TET3) in combination with Piezo2 knockdown to see whether fibroblast activation/myofibroblast differentiation would be rescued by Piezo2 knockdown. I would suggest that the authors tone down the statement about the TET3-Piezo2 axis.
2. Have the Col1a2-CreERT2 mice the authors used in the paper been previously confirmed to have cardiac fibroblast-specific TET3 knockout, given that no positive reporter signal was detected in cardiac fibroblasts derived from the Col1a1-CreERT2 mice (LP Aguado-Alvaro et al, *Biomedicines*, 2022, 10:2350)? If the Col1a2-CreERT2 mice did not have cardiac fibroblast-specific TET3 knockout, the therapeutic effects associated with these mice would not be related to the function of TET3 in unactivated cardiac fibroblasts. In addition, the citation (ref 2 in Supplementary Methods) for the Col1a2-CreERT2 mice was incorrect. Ref 2 mentioned the Col1a1-CreERT2 mice but not the Col1a2-CreERT2 mice.

Minor points:

1. The paper used Bobcat339 in vivo but the source, route of injection (i.p., i.v. or s.c.) and dosage were not specified.

2. As Bobcat339 is a well-established TET3 protein degrader and not an enzymatic inhibitor (Lv et al, PNAS, 2023; Lv et al, JCI, 2024), I would suggest that on page 5 of the manuscript, the authors clearly describe Bobcat339 as a TET3 degrader instead of an enzymatic inhibitor. The statement on page 5 "Bobcat339 is a small-molecule TET3 degrader that has been reported to inhibit TET3 activity in different settings" was confusing.

Referee #1 (Comments on Novelty/Model System for Author):

This study is interesting. The authors demonstrate the involvement of TET3-Piezo2 axis in the progression of cardiac fibrosis using fibroblast-specific TET3 and Piezo2 deficient mice and already known TET3 degrading small molecule, Bobcat339. They also suggest the involvement of TET3-Piezo2 axis in patients with heart failure. This paper is well organized, and the data are solid and concretely demonstrated.

Referee #1 (Remarks for Author):

This study is interesting. The authors demonstrate the involvement of TET3-Piezo2 axis in the progression of cardiac fibrosis using fibroblast-specific TET3 and Piezo2 deficient mice and already known TET3 degrading small molecule, Bobcat339. They also suggest the involvement of TET3-Piezo2 axis in patients with heart failure. This paper is well organized, and the data are solid and concretely demonstrated. However, there are concerns on data **EMM-2025-21496** presentations.

We thank the reviewer for his/her insightful comments and constructive critiques. Please see below our point-by-point response.

Comments:**1. Scale bar should be shown, e.g., Figure 1B-D, G-I.**

We thank the reviewer for his/her comment. Scale bar has been added for Figures 1B, 1C, 1D, 1E, 1G, 1H, 1I, 1J, 3B, 3C, 3D, 3E, 5B, 5C, 5D, 5E, S1B, S1C, S1D, S2B, S2C, S2D, S7B, S7C, S7D, S12B, S12C, S12D, S12E, S14B, S14C, S14D, and S14E.

2. It is curious for me why background color is so high in Ad-GFP TGF- β image compared with other groups. The background color, especially non-cell area, should be the same among 4 groups.

We thank the reviewer for his/her comment. We have replaced the original images shown in Figure 1I with the ones of even background.

3. Figure 2F, 2P, 3K and 4K. Why do they show only two groups? Is it sham groups or TAC-treated groups? It is sufficient to show either picosirius red images or masson trichrome images, but they need to show all 4 groups.

We thank the reviewer for his/her suggestion. We have provided PSR/Masson's staining for the sham groups shown in Figures 2F, 2P, S4B, and S5B. Please note that no sham groups were included in the experiments shown in Figure 3 and Figure 5 and all the mice subjected to the TAC surgery.

4. Figure 3. I am wondering if the concentration (50-100 μ M) of Bobcat339 is high. Is this reagent less selective? If so, they need to confirm whether the pharmacological action of Bobcat339 is associated with TET3 degradation.

We appreciate the reviewer's concern. We agree with the reviewer that compared to the

dose (10 μ M) reported in macrophages (PMID: 39141428) and in hypothalamic neuronal cells (PMID: 37036983), the dose used in our study appears to be higher. We perform a titration of Bobcat339 in both human and murine fibroblasts to show that the extent to which TET3 was degraded correlated with the extent to which myofibroblast activation was inhibited. It should be noted that the original paper (PMID: 35586434) that ruled out TET1 or TET2 as target for Bobcat339 used a dose (50 μ M) similar to ours.

Referee #2 (Comments on Novelty/Model System for Author):

Authors showed that the TET3-Piezo2 pathway in fibroblasts promotes cardiac fibrosis. In vitro experiments using fibroblasts, and in vivo experiments using fibroblast-specific knockout mice and TET3 degrader (Bobcat339) all strongly indicate the involvement of TET3-Piezo2 in fibrosis. The results obtained are consistent between figures, making the results convincing. However, there is a concern. The heterogeneity has been reported in fibroblasts. It would be better to mention in which type of fibroblasts TET3 is expressed.

Referee #2 (Remarks for Author):

In this study, authors showed that TET3 is involved in fibrosis by promoting the conversion of fibroblasts to myofibroblasts by controlling the expression of Piezo2. In vitro and in vivo experiments have shown consistent results of the importance of the TET3-Piezo2 pathway. In vivo, the effect of Bobcat339, a TET3-specific degrader, was also observed, which is interesting from a clinical side. However, in recent years, heterogeneity has been reported in fibroblasts, so I think it is necessary to mention in which type of cells TET3 is expressed. Overall, they have provided sufficient data, and no further experiments are necessary.

We thank the reviewer for his/her insightful comments and constructive critiques and for his/her enthusiastic support. We have since analyzed the expression pattern of TET3 in different populations of cardiac fibroblasts using a previously published single-cell RNA-seq dataset (PMID: PMID: 33839759). As shown in the new Figure S3, TET3 seems to be preferentially expressed in cluster 1 cardiac fibroblasts, which are annotated to mediate “response to stimuli”.

Referee #3 (Comments on Novelty/Model System for Author):

It is not clear whether the Col1a2-CreERT2 mice used in the study had cardiac fibroblast-specific Tet3 ablation.

Referee #3 (Remarks for Author):

RE: EMM-2025-21496

Using siRNA knockdown in mouse and human primary cardiac fibroblasts the authors showed that TET3 was required for fibroblast activation and myofibroblast differentiation induced by profibrotic stimuli such as TGF- β 1. In two different murine models of cardiomyopathy (TAC and LAD), the authors demonstrated that fibroblast (Col1a2-CreERT2)- or myofibroblast (Postn-CreERT2)-specific Tet3 ablation mitigated cardiac fibrosis and improved heart function and that the effects were mimicked by Bobcat339, a small molecule TET3-specific degrader. To investigate the underlying mechanisms, the authors performed Cre-mediated TET3 knockdown in cardiac fibroblasts followed by integrated genome-wide analyses, RNA-seq and CUT&Tag-seq, and identified genes and pathways affected by TET3.

Among the 68 putative TET3 targets identified, the authors focused on Piezo2, a mechanosensitive ion channel protein previously implicated in dermal fibrosis and cardiac fibrosis by other research groups. The authors provided in vitro evidence in cardiac fibroblasts showing that TET3 positively regulated Piezo2 transcription at least in part by binding to the Piezo2 gene promoter inducing 5hmC modification. To address the in vivo role of Piezo2, the authors used a Piezo2 inhibitor, GsTMx4, and observed therapeutic effects in two mouse models of cardiac fibrosis. To find out how Piezo2 might regulate fibroblast activation, the authors performed ATAC-seq and CUT&Tag-seq analyses in cardiac fibroblasts and observed decreased chromatin association of pro-fibrogenic transcription factors SRF and TEAD following Piezo2 knockdown. Finally, the authors revealed a positive correlation between the expression of TET3 and Piezo2 in human heart tissues as well as a higher expression of TET3 and Piezo2 in heart tissues of patients with heart failure as compared to healthy controls. The authors concluded that TET3 plays an important role in the pathogenesis of cardiac fibrosis and may be potentially targeted for therapy.

Overall, the paper was well-written and data were clearly presented. The findings that TET3 in cardiac fibroblasts contributes critically to myocardial fibrosis and that TET3 may serve as a potential therapeutic target are novel and important for the cardiac research field. In addition, the genome-wide data generated from the study provide mechanistic insights onto how TET3 regulates cardiac fibroblasts in the context of myocardial fibrosis, and, if made publicly available, will be a rich source for the research community.

We thank the reviewer for his/her insightful comments and constructive critiques. The RNA-seq and CUT&Tag-seq data have been uploaded to PubMed and are currently pending

administrative approval. These data will become publically available as soon as the manuscript is published. Please see below our point-by-point response.

Major points:

1. The importance of the TET3-Piezo2 axis in cardiac fibrosis is not well supported. The authors showed in cardiac fibroblasts that TET3 regulated Piezo2 transcription but did not provide further evidence that Piezo2 is a major downstream effector of TET3, especially given the multitude of potential TET3 targets identified by RNA-seq and CUT&Tag-seq. The authors showed partial therapeutic effects using a Piezo2 inhibitor (GsTMx4). However, as GsTMx4 is not a Piezo2-specific inhibitor and which also inhibits other cationic mechanosensitive channels such as Piezo1 and TRP channels, the in vivo role of Piezo2 as a major downstream effector of TET3 cannot be established. Perhaps, the authors can overexpress TET3 in cardiac fibroblasts using viral vectors (not by TGF- β 1 stimulation which will alter gene expression in addition to expression of TET3) in combination with Piezo2 knockdown to see whether fibroblast activation/myofibroblast differentiation would be rescued by Piezo2 knockdown. I would suggest that the authors tone down the statement about the TET3-Piezo2 axis.

We appreciate the reviewer's concern. We recognize that the current evidence is insufficient to support a "TET3-Piezo2" axis. We have made appropriate changes throughout the manuscript to avoid such an overstatement. We have also added a few sentences in the Discussion section to explicitly state that the more evidence is needed to implicate Piezo2 as a genuine regulator of cardiac fibrosis and heart failure.

2. Have the Col1a2-CreERT2 mice the authors used in the paper been previously confirmed to have cardiac fibroblast-specific TET3 knockout, given that no positive reporter signal was detected in cardiac fibroblasts derived from the Col1a1-CreERT2 mice (LP Aguado-Alvaro et al, Biomedicines, 2022, 10:2350)? If the Col1a2-CreERT2 mice did not have cardiac fibroblast-specific TET3 knockout, the therapeutic effects associated with these mice would not be related to the function of TET3 in unactivated cardiac fibroblasts. In addition, the citation (ref 2 in Supplementary Methods) for the Col1a2-CreERT2 mice was incorrect. Ref 2 mentioned the Col1a1-CreERT2 mice but not the Col1a2-CreERT2 mice. We appreciate the reviewer's concern. The Col1a2-Cre-ERT2 strain has been widely used as a driver for fibroblast-specific gene targeting including in the heart (PMID: 24899689, 25317562, 27867037, 29160304, 29601781, 33500351, 34132780, 34324438, 36438502, 37227779, 37437652, 38503742, 38682176). That being said, we recognize that there are now other Cre drivers (e.g., Tcf21-MCM) that offer increased specificity and efficiency for gene targeting in quiescent cardiac fibroblasts. We have added a few sentences in the Discussion section to explicitly spell out the limitation. We have also corrected the reference for the Col1a2-CreERT2 mice and we apologize for the confusion.

Minor points:

1. The paper used Bobcat339 in vivo but the source, route of injection (i.p., i.v. or s.c.) and dosage were not specified.

We thank the reviewer for his/her suggestion. The route whereby Bobcat339 was injected

(i.p.) is now clearly stated in the Methods section.

2. As Bobcat339 is a well-established TET3 protein degrader and not an enzymatic inhibitor (Lv et al, PNAS, 2023; Lv et al, JCI, 2024), I would suggest that on page 5 of the manuscript, the authors clearly describe Bobcat339 as a TET3 degrader instead of an enzymatic inhibitor. The statement on page 5 "Bobcat339 is a small-molecule TET3 degrader that has been reported to inhibit TET3 activity in different settings" was confusing.

We thank the reviewer for his/her suggestion. We have rephrased the sentence referred to by the reviewer to avoid confusion.

7th Aug 2025

Dear Dr. Kong,

Thank you for the submission of your revised manuscript to EMBO Molecular Medicine. We have now received the enclosed report from the referee who was asked to re-assess it. As you will see, the referee is now supportive, and I am pleased to inform you that we will be able to accept your manuscript pending the following amendments:

On a more editorial level:

1. Please remove the Authors' Contribution section from the manuscript text.

2. Data availability: It is mandatory to include a 'Data Availability' section. Before submitting your revision, primary datasets produced in this study (such as the RNA-seq data and CUT&Tag-seq data) need to be deposited in an appropriate public database, and the accession numbers and database listed under 'Data Availability'.

More information can be found here: <https://www.embopress.org/page/journal/17574684/authorguide#dataavailability>.

3. Please ensure that the funding information provided in the submission system matches the details in the manuscript text. Currently, grant numbers 82400729 and 82370633 are listed in the manuscript but are missing from the submission system. This discrepancy must be corrected.

4. The section titled "Conflicts of Interest" should be renamed to "Disclosure and competing interests statement." Additionally, please remove the word "none" and replace it with the following phrasing: The authors declare no competing interests.

5. Please download and fill our Reagents and Tools Table template (.docx), which you can find in our author guidelines: <https://www.embopress.org/page/journal/17574684/authorguide#structuredmethods>

6. Please add the missing callouts for Figure 2K and Appendix Table S3.

7. Appendix

- Rename the supplementary tables and figures file to "Appendix" and upload it as a single PDF file.

- Move the Supplementary Methods section into the main manuscript and integrate it with the Methods section.

- Include a Table of Contents with page numbers in the Appendix file.

- Update the nomenclature throughout the Appendix and manuscript text to follow the format: Appendix Figure S1, Appendix Table S1, etc.

8. Source data

- Please upload a completed Source Data Checklist.

- Source data for all main figures should be uploaded as one file per figure.

- Source data for appendix figures should be compiled together and uploaded as a single file.

- Images should not be submitted in .ppt format; we recommend using .tiff format

9. During a routine image analysis, we noticed a potential duplication between Figure 3E (vehicle, lower panel) and Figure S13 (Ad-GFP, lower panel). Please verify whether these are the same images or simply appear highly similar.

10. The paper explained : This section should be revised to follow a structured format, including three separate subsections: "Problem," "Results," and "Impact."

Please refer to any of our published articles for examples of the correct format and content.

11. Please provide a 'Synopsis' to further enhance discoverability. Synopses are displayed on the journal webpage and are freely accessible to all readers. They include a short stand first (maximum of 300 characters, including space) as well as 2-5 one-sentences bullet points that summarizes the paper. Please write the bullet points to summarize the key NEW findings. They should be designed to be complementary to the abstract - i.e. not repeat the same text. We encourage inclusion of key acronyms and quantitative information (maximum of 30 words / bullet point). Please use the passive voice. Please attach these

in a separate file or send them by email, we will incorporate them accordingly.

Please also suggest a visual abstract to illustrate your article as a PNG file 550 px wide x 300-600 px high.

12. Please address the following issues related to figure legends:

- Please note that the exact p values are not provided in the legends of figures 1A, B; 2G, H, I, J, Q, R, S, T; 3A, L, M, N, O; 4M, O, U, Q, S, V; 5A, L, M, N, O
- Please indicate the statistical test used for data analysis in the legends of figures 6B, C, D, G
- Please note that the error bars are not defined in the legends of figures 2B, C, D, E, G, H, I, J, L, M, N, O, Q, R, S, T; 7A

13. Please correct the order and headings of the manuscript sections to: Abstract / Keywords / The Paper Explained / Introduction / Results / Discussion / Methods / Data Availability / Acknowledgements / Disclosure and Competing Interests Statement / References / Main Figure Legends / Tables / Expanded View Figure Legends

I look forward to seeing a revised form of your manuscript as soon as possible.

Yours sincerely,
Jingyi

Jingyi Hou
Senior Editor
EMBO Molecular Medicine

*** Instructions to submit your revised manuscript ***

***** Reviewer's comments *****

Referee #3 (Remarks for Author):

The revised manuscript has been significantly improved over the initial version, I have no further comments.

The authors addressed the remaining editorial issues.

18th Aug 2025

Dear Dr. Kong,

We are pleased to inform you that your manuscript is accepted for publication and is now being sent to our publisher to be included in the next available issue of EMBO Molecular Medicine.

Sincerely,
Jingyi

Jingyi Hou
Senior Editor
EMBO Molecular Medicine
